# Ancestral functionality and symbiotic refinement of NIN in root nodule symbiosis

Jieyu Liu [1,6], Siqi Yan [1,6], Min Li[1,2], Defeng Shen [1,4], Michaela Tichá [1,5], René Bærentsen[3], Kasper Røjkjær Andersen [3], Floris Verbeek[1], Olga Kulikova[1], Rene Geurts [1] ✉, Ton Bisseling [1,2,7] ✉ & Rik Huisman[1,7] ✉

Nitrogen-fixing nodule symbiosis is an ecologically and economically important trait in legumes and some related species. A critical step in the evolution of nodulation is the recruitment of NODULE INCEPTION (NIN); a homolog of the nitrate-sensing NIN-LIKE PROTEIN (NLP) transcription factors. However, whether adaptations have occurred in the NIN protein upon its recruitment in symbiosis remains elusive. Here we show that non-symbiotic NIN orthologs can function in intracellular infection and even nodule initiation, indicating that these properties of NIN predate the evolution of nodulation. Concurrent with the evolution of nodulation, symbiotic NIN proteins were optimized for their role in symbiosis by acquiring nitrate independent functionality, including constitutive nuclear localization. A single amino acid substitution in the non-symbiotic Arabidopsis AtNLP2 enhances its nuclear localization under low nitrate conditions, making it functionally comparable to the symbiotic Parasponia PanNIN. Our study provides insight in the evolutionary trajectory and molecular adaptation that allowed NIN to function as the central regulator of nitrogen-fixing nodule symbiosis.

Some plant species can establish a nitrogen-fixing root nodule endosymbiosis with diazotrophic rhizobium or *Frankia* bacteria. Nodulating species are members of the related orders Fabales, Fagales, Cucurbitales, and Rosales, known collectively as the nitrogen-fixing clade (NFC)[1,2]. The evolution of nodulation depends on at least two critical events that occurred in the common ancestor of the NFC[3]. First, the recruitment of the common symbiotic signaling pathway from the more ancient arbuscular mycorrhizal endosymbiosis, which is widespread across the plant kingdom[4]. This signaling initiates both endosymbiotic interactions upon detecting a symbiotic microbe. Second, the recruitment of the transcription factor nodule inception (NIN). In the common ancestor of the NFC, *NIN* gained a *cis*-regulatory element in its promoter, placing its expression directly under the control of the

common symbiotic signaling pathway[5]. As the most upstream nodulation-specific transcription factor, NIN functions as the master regulator essential for multiple steps of nodule formation and functioning[6–13]. However, whether adaptations in the NIN protein itself were critical to commit to such a key function in nodulation remains elusive.

NIN is part of the NIN-like protein (NLP) family[14]. NLPs are primary nitrate sensors, playing a central role in nitrate-induced signaling[15]. The *NIN* orthogroup arose upon a duplication in an ancestral dicot plant species, predating the evolution of the NFC[16]. Therefore, nonnodulating plant species outside the NFC have *NIN* orthologs. NIN orthologs from within and outside the NFC induce distinct downstream genes. Symbiotic NINs induce genes involved in nodule

[1]Laboratory of Molecular Biology, Department of Plant Sciences, Wageningen University & Research, Wageningen, The Netherlands. [2]Key Lab of Grassland Resources of the Ministry of Education of China, College of Grassland Science, Inner Mongolia Agricultural University, Hohhot, China. [3]Department of Molecular Biology and Genetics, Aarhus University, Aarhus, Denmark. [4]Present address: Max Planck Institute for Plant Breeding Research, Cologne, Germany. [5]Present address: Department of Biology, Faculty of Natural Sciences, Norwegian University of Science and Technology, Trondheim, Norway. [6]These authors contributed equally: Jieyu Liu, Siqi Yan. [9]These authors jointly supervised this work: Ton Bisseling, Rik Huisman ✉e-mail: rene.geurts@wur.nl; ton.bisseling@wur.nl; rik.huisman@wur.nl

development, such as *NF-YA* and *NF-YB* encoding subunits of the nuclear factor Y complex[17], while nonsymbiotic NIN orthologs like *Arabidopsis thaliana* AtNLP2 regulate nitrate-responsive genes[18]. Moreover, the subcellular localization of symbiotic and nonsymbiotic NIN proteins is different. Symbiotic NINs are nuclear localized[10,19], whereas nonsymbiotic NINs are generally cytoplasmic under low nitrate conditions and only shuttle to the nucleus upon sensing high nitrate[18,20]. This suggests that changes also occurred in the NIN protein when it was recruited in nodulation.

Here, we show that nonsymbiotic NIN orthologs from several species can function in intracellular infection and even nodule initiation, albeit with low efficiency. Further analyses reveal that the major functional improvement is not in the DNA binding ability but nitrate-independent nuclear localization and functioning.

## Results

### Nonsymbiotic NIN orthologs are partially functional in nodulation

To test to what extent nonsymbiotic NIN proteins can function in nodulation, we selected orthologs representing a range of phylogenetic distances to the legume experimental model *Medicago* (*Medicago truncatula*). These include *PanNIN* of the nodulating nonlegume species *Parasponia* (*Parasponia andersonii*), and the nonsymbiotic orthologs of three species outside the NFC; *MeNIN1* and *MeNIN2* of cassava (*Manihot esculenta*), *AtNLP1*, *AtNLP2*, and *AtNLP3* of *Arabidopsis*, and *SlNIN* of tomato (*Solanum lycopersicum*)[14,19,21] (Fig. 1a). In addition, the *Medicago NIN* paralog *MtNLP1* was included. We used these NIN homologs to trans-complement the *Medicago Mtnin-1* knock-out mutant, which can neither form infection threads nor nodules[9]. The experiments were conducted under 0.5 mM nitrate conditions. As codon usage in *NIN* orthologs of the different plant species is similar (Supplementary Data 1), we used native coding sequences driven by the *Medicago MtNIN* promoter[22]. These trans-complementation experiments showed that nodules were efficiently formed on roots transformed with symbiotic *NIN* genes *MtNIN* and *PanNIN* (Fig. 1b). Notably, NIN orthologues from three non-NFC species; *Arabidopsis* (AtNLP2), cassava (MeNIN1) and tomato (SlNIN), were also able to restore infection thread formation (Fig. 1b, k–m, q, r). In contrast, *MtNLP1* was unable to restore infection thread formation (Fig. 1p), indicating a larger functional divergence in the NIN paralogue than in the orthologues. When trans-complementing *Mtnin-1* with *AtNLP2* also some infected nodule primordia were observed, whereas other nonsymbiotic NINs and MtNLP1 did not induce such structures (Fig. 1b and Supplementary Fig. 1). These results show that all three nonsymbiotic species have a NIN orthologue that can restore infection thread formation in *Mtnin-1*, while AtNLP1, AtNLP3, and MeNIN2 have lost this ability, likely due to neo-functionalisation after lineage-specific duplication.

The nodules formed by native MtNIN were pink (Fig. 1c), indicating the accumulation of leghemoglobin, a hallmark of functional nitrogen-fixing nodules. Sections of these nodules showed that they have a wildtype zonation. The rhizobium bacteria in the cells of the fixation zone were fully differentiated/elongated, like in wildtype nodules (Fig. 1d, e). In contrast, nodules formed on roots complemented with *PanNIN* were white. While bacteria were released into cells of these nodules, they did not fully differentiate, and premature senescence occurred in the basal part of the nodules (Fig. 1f–h). The *Parasponia* and *Medicago* lineages diverged early in the NFC[1]. The lack of full trans-complementation of *Mtnin-1* by *PanNIN* suggests that, after the recruitment of NIN by the common ancestor of the NFC, lineage-specific evolution still occurred. The differentiation and maintenance of bacteroids in *Medicago* nodules after release could depend on adaptations in MtNIN that are not present in PanNIN. As in *Parasponia* nodules, rhizobium bacteria are not released and elongated, PanNIN would not require these adaptations. The nodule

primordia formed on roots complemented with *AtNLP2* were arrested at an early stage of development (Fig. 1i–k). Although infection threads penetrated the nodule cells, rhizobia were not released. These findings suggest that NIN possessed ancestral symbiotic functionality and subsequently experienced lineage-specific adaptations that optimized its function in nodule formation.

To further investigate the evolution of NIN, we used ancestral sequence reconstruction, a technique to infer the most probable sequence of internal nodes of a phylogenetic tree of a protein family. These sequences are an approximation of the ancestral states of the protein family during evolution[23,24]. For this, we analyzed the protein sequence of a wide range of dicot NIN orthologs, paralogs from the NIN sister clade containing MtNLP1, as well as monocot NIN/NLP1 homologs that predate the split of the NIN and NLP1 orthogroups (Supplementary Data 4). We used two independent ancestral sequence reconstruction algorithms, MrBayes[25] and GRASP[24], followed by manual curation. With this approach, we predicted the NIN protein sequences of the most recent ancestor of the nitrogen-fixing clade ($NIN_{NFC}$) and the most recent nonsymbiotic ancestor of the Rosid clade ($NIN_{Rosids}$) (Supplementary Fig. 7 and Supplementary Data 2). Although the presence of multiple non-conserved regions in NIN poses challenges for reliable ancestral sequence reconstruction, the structure of $NIN_{Rosids}$ resembles that of extant NIN proteins (Supplementary Fig. 2). We used gene synthesis to generate *Medicago* codon-optimized DNA sequences to functionally characterize the predicted ancestral proteins. Introduction of $NIN_{Rosids}$ in *Mtnin-1* mutant roots resulted in infection thread formation (Supplementary Fig. 2), supporting a latent capacity for symbiotic functioning of the ancestral Rosid NIN. We next asked whether the ability to initiate nodule primordia was already present in NIN of the NFC ancestor ($NIN_{NFC}$). To test this, we resurrected $NIN_{NFC}$ and used it to complement *Mtnin-1*. Besides infection threads, this resulted in rare but observable nodule primordia (Supplementary Fig. 2).

Taken together, these results show that *NIN* orthologs from non-nodulating species can partially function in nodulation, and the ability of NIN to induce nodule primordia was likely already present in the NFC ancestor, being at the basis of the evolution of nodulation.

### NIN recruitment in nodulation did not require major changes in the C-terminal DNA-binding domain

Since nonsymbiotic NIN proteins function with markedly lower efficiency compared to symbiotic NINs, changes likely occurred in the NIN protein sequence that contributed to its functioning in nodulation. As NIN proteins function as transcription factors, we first tested the ability of NIN orthologs to target symbiotic genes. In *Lotus japonicus*, the C-terminus of NIN, containing its DNA-binding domain, binds *cis*-regulatory elements of *LjNF-YA1* and *LjNF-YB1*. In contrast, the NIN paralog *LjNLP4* does not bind these sites[26]. Thus, changes in the NIN C-terminus may enable it to regulate symbiotic genes. NIN-binding sites were identified in symbiotic NIN-targets *MtLBD16*, *MtCEP7*, *MtCLE13*, as well as the *Medicago* orthologs of *LjNF-YA1* and *LjNF-YB1* (*MtNF-YA1*, *MtNF-YB16*)[27–29] (Supplementary Fig. 3d). These sites resemble those bound by the nonsymbiotic ortholog AtNLP2[18]. Using electrophoretic mobility shift assays (EMSA), we observed significant DNA mobility shifts after incubation with the C-terminus of MtNIN, PanNIN, MeNIN1, MeNIN2, AtNLP2, and SlNIN, indicating that these proteins can bind symbiotic NIN-binding sites in vitro (Fig. 2a and Supplementary Fig. 3a, b).

To test whether in vitro binding corresponds to in vivo ability to activate gene expression, we generated constructs in which the N-terminus of MtNIN was fused to the C-terminus of the tested NIN orthologs and the glucocorticoid receptor (GR), allowing dexamethasone-controlled nuclear localization of the fusion proteins (Supplementary Fig. 4b). *Medicago* plants producing these chimeric fusion proteins were treated with dexamethasone for 16 h. Following

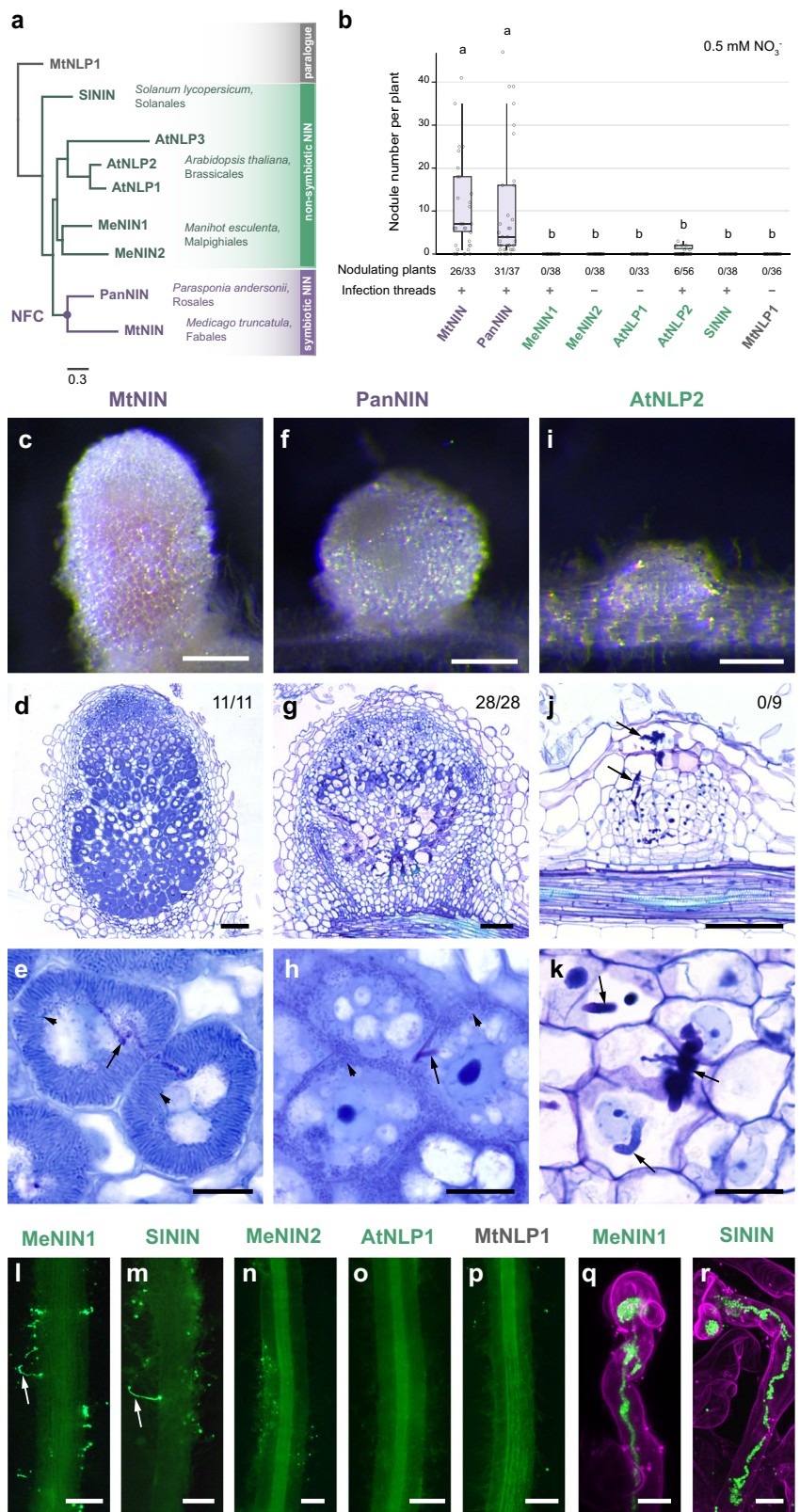

treatment, symbiotic NIN targets were induced by the symbiotic MtNIN and the PanNIN chimer, as well as by chimers containing the nonsymbiotic MeNIN1, AtNLP2, and SlNIN C-termini (Fig. 2b and Supplementary Fig. 4a). The induction of the tested NIN targets by the symbiotic NIN-chimers was generally strong, while the induction by nonsymbiotic NIN-chimers was variable depending on the target gene. Most nonsymbiotic NIN orthologs could induce some of these symbiotic NIN targets, but none of them significantly induced all targets. Similar results were obtained when using the full-length proteins (Supplementary Fig. 5).

To test whether the ability to induce symbiotic NIN targets is sufficient for functionality in symbiosis, we introduced chimeric NIN proteins (Supplementary Fig. 4c) into the *Mtnin-1* mutant roots. Nodules were formed on mutant plant roots complemented with

**Fig. 1 | NIN orthologs from different lineages are partially functional in nodule symbiosis. a** Phylogenetic tree of studied NIN orthologs and paralog MtNLP1. NFC nitrogen-fixing clade. Mt *Medicago truncatula*, Pan *Parasponia andersonii*, Me *Manihot esculenta*, At *Arabidopsis thaliana*, Sl *Solanum lycopersicum*. **b** Number of nodules formed on Mtnin-1 mutant roots complemented with different NIN orthologs and a paralog. Plants were harvested at 4 weeks post inoculation with *Sinorhizobium meliloti* 2011 expressing GFP. Box plots show the median (center line) and interquartile range (box) of the number of nodules per nodulating plant; points represent individual observations. Lowercase letters indicate significant differences between samples (Kruskal–Wallis and post-hoc Dunn's test, Benjamini–Yekutieli adjusted $p < 0.05$). Purple: symbiotic NIN, green: nonsymbiotic NIN, gray: NIN paralog. The order of images from left to right reflects the

phylogenetic distance to *Medicago*. Source data are provided as a Source data file. **c–k** Images of nodules formed on Mtnin-1 complemented with MtNIN, PanNIN, and AtNLP2. **c, f, i** Stereomicroscope images. Scale bars: 2 mm. **d, g, j** Longitudinal sections stained with toluidine blue. Numbers indicate nodules with released bacteria. Scale bars: 100 μm. **e, h, k** Magnification of nodule cells. Arrows indicate infection threads; arrowheads indicate released rhizobia. Scale bars: 20 μm. **l–p** Green fluorescence stereomicroscopy images showing infection threads (arrows) formed on the roots transformed with MeNIN1 and SlNIN. Scale bars: 2 mm. **q, r** Confocal images of transgenic roots stained with propidium iodide (magenta) showing infection threads (green) formed in the root hairs. Scale bars: 10 μm.

chimeric proteins containing the C-terminus of symbiotic NINs and nonsymbiotic NINs, MeNIN1, MeNIN2, AtNLP2, and SlNIN (Fig. 2c, d). On these roots, nodules were formed with phenotypes ranging from infected primordia (MeNIN2), release of rhizobia in nodule cells (MeNIN1), disorganized but differentiated rhizobia (AtNLP2), to radially organized differentiated rhizobia in pink nodules (SlNIN) (Fig. 2d). Neither nodules nor infection threads were formed on roots transformed with the empty vector control or constructs including the C-terminus of AtNLP1 or MtNLP1.

Taken together, these results show that the C-terminus of most tested nonsymbiotic NIN proteins allows activation of symbiotic targets and is able to function in nodulation. The different levels of symbiotic functionality of the full-length proteins (Fig. 1) do not always correlate with their phylogenetic distance to *Medicago*, their DNA binding efficiency and transactivation of symbiotic NIN-targets (Fig. 2). These data imply that the functional difference of NIN orthologs in nodulation, including the release and differentiation of rhizobia, is mainly associated with their N-terminal sequence.

## Nuclear localization is required for nonsymbiotic NIN to function in a symbiotic context

Using the N-terminus from MtNIN enabled most nonsymbiotic NINs to induce nodules (Fig. 2), strongly suggesting that the major functional improvements for nodulation occurred in the N-terminus of NIN. Previous studies on the N-terminus of AtNLP7 showed that it has a nitrate-sensing domain and a phosphorylation site that is required for nitrate-triggered nuclear retention[30–32]. To test whether nitrate-dependent nuclear localization was changed when NIN was recruited in nodulation, we determined the subcellular localization of NIN orthologs in *Medicago* nodules under 0 mM and 20 mM nitrate. For this, NIN orthologs and paralog MtNLP1 fused to GFP were expressed in a wildtype background. Rhizobium-inoculated *Medicago* plants were grown in the absence of nitrate for 4 weeks. Nodules of these plants were used for subcellular localization studies; first in the absence of nitrate and subsequently after 1 h of 20 mM KNO$_3$ treatment (Fig. 3a). We quantified the ratio of GFP intensity in nucleus and cytoplasm under both conditions to visualize nitrate-dependent subcellular localization (Fig. 3b). This showed that symbiotic MtNIN and PanNIN were nuclear localized under both zero and high nitrate conditions. In contrast, AtNLP2, SlNIN, and MtNLP1 were localized in the cytoplasm and shuttled to the nucleus upon high nitrate application, consistent with previous studies[10,18,33]. Similarly, MeNIN1 shuttled to the nucleus upon nitrate application, albeit with lower efficiency. The subcellular localization of AtNLP1 and MeNIN2 was nitrate independent; AtNLP1 remained in the cytoplasm, whereas MeNIN2 was nuclear localized in both conditions (Fig. 3a, b). MeNIN2 might function similarly as what has been reported for AtNLP8, which is constitutively nuclear localized but requires nitrate to activate downstream gene expression[34]. Taken together, these data show that the nuclear localization of most nonsymbiotic NINs is nitrate dependent, whereas symbiotic MtNIN and PanNIN are constitutively nuclear localized.

Despite AtNLP2 being nitrate-dependent for nuclear localization, it was still able to partially function in symbiosis. This could be due to the presence of a low amount of exogenous nitrate (0.5 mM) in this *trans*-complementation experiment (Fig. 1). To test whether the function of AtNLP2 in nodulation is nitrate-dependent, we examined the symbiotic functioning of AtNLP2 in the absence of nitrate. *Mtnin-1* roots complemented with *AtNLP2* formed nodules under low nitrate (0.5 mM), but in the absence of nitrate, neither nodules nor infection threads were observed (Fig. 3c). This correlated with a lower amount of AtNLP2 in the nucleus under these conditions (Fig. 3d, e). Similar results were obtained for MeNIN1 ($n = 22$), as no infection threads were observed in the absence of nitrate. In contrast, *Mtnin-1* roots transformed with *MtNIN* formed nodules in the absence of nitrate (Supplementary Fig. 6a). To assess whether enhanced nuclear localization could improve AtNLP2 function, we fused nuclear localization signals (NLS) to its termini and expressed these in *Mtnin-1*. Despite increased nuclear accumulation, these constructs failed to induce nodules (Supplementary Fig. 6b–d), likely due to interference of the NLS, as similar fusions to MtNIN also impaired its function (Supplementary Fig. 6e). These data show that nonsymbiotic NIN, such as AtNLP2, requires nitrate to be nuclear localized and function in nodulation.

## A single amino acid adaptation makes AtNLP2 functionally similar to a symbiotic NIN

Previous studies showed that a specific phosphorylation site required for nitrate-triggered nuclear retention has been lost in legume NIN proteins[31,32,35,36]. Sequence analysis indicated that this is the result of lineage-specific events that occurred multiple times after NIN was recruited in nodulation, while it was maintained in multiple nodulating nonlegume species, such as *Parasponia* (Supplementary Fig. 7). Thus, loss of this phosphorylation site was not a pre-requisite for NIN to be recruited in root nodule symbiosis.

To identify changes enhancing symbiotic NIN function, we compared symbiotic NINs with nonsymbiotic NIN orthologs across a wide range of plant species and identified seven residues specifically conserved in symbiotic NINs, suggesting positive selection in the NFC (Fig. 4a, Supplementary Fig. 7, and Supplementary Data 2). Among the identified motifs, positions 1 and 5 have the potential to create an extra NLS for symbiotic NINs (Fig. 4a), whereas position 2, also identified by Zhang et al.[35], is in the middle of the nitrate binding domain[30]. Positions 5, 6, and 7 are relatively close to the DNA-binding domain. Positions 4, 5, and 6 are within a region for which no specific structure is predicted (Fig. 4a, b).

To test whether the identified amino acid changes represent important motifs for symbiotic NIN functioning, we introduced the corresponding AtNLP2 residues into MtNIN. This showed a significantly lower number of nodules formed by the mutated MtNIN$^{M1–7}$ compared to MtNIN control, although the fewer nodules induced by MtNIN$^{M1–7}$ were still functional (Fig. 4c and Supplementary Fig. 8). This implies that one or more of these seven residues are important for nodule formation.

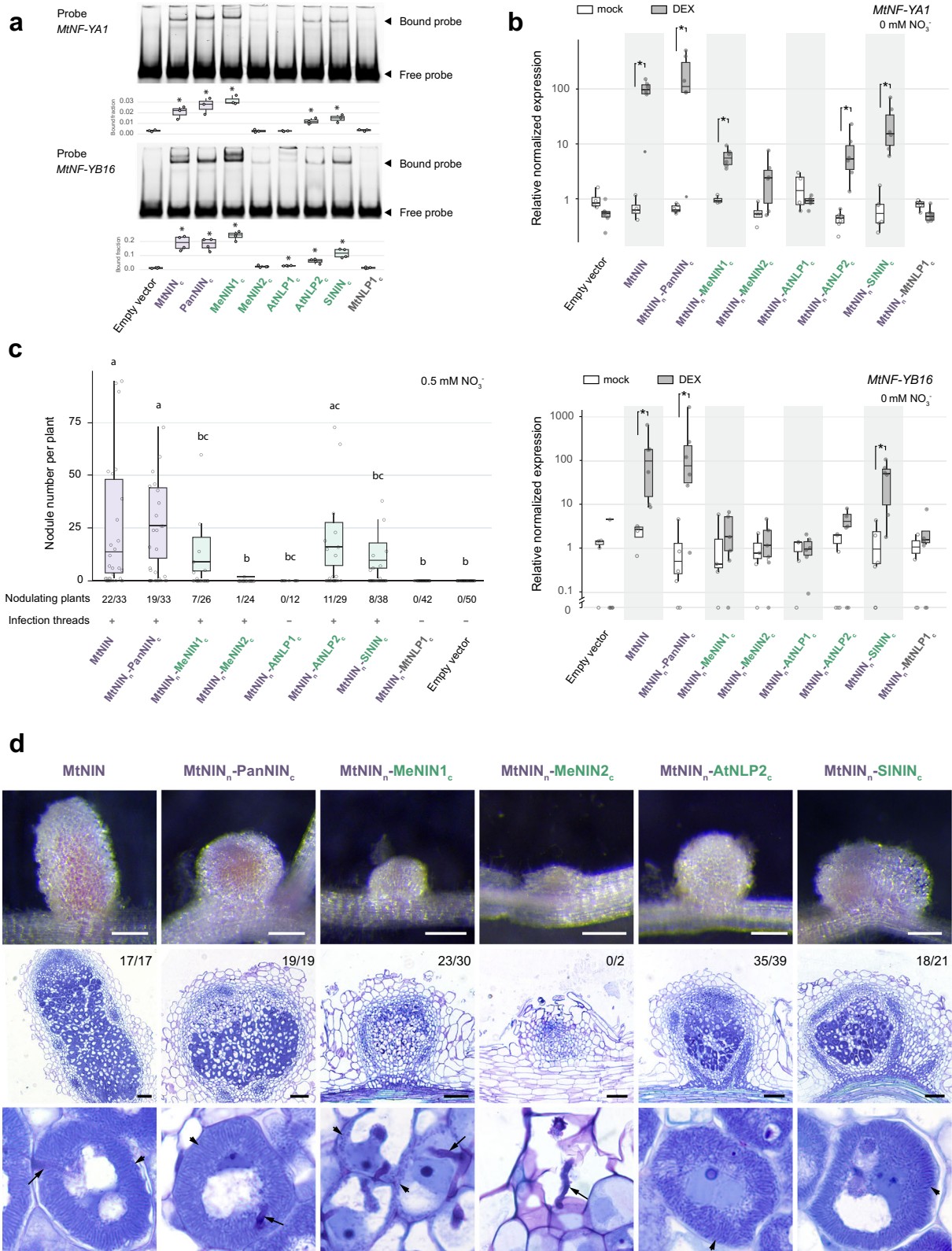

To study whether single sites of amino acid changes can improve the function of nonsymbiotic NIN, we introduced each of seven symbiotic-NIN-specific residues into AtNLP2 and assessed their function in trans-complementation of *Mtnin-1*. Compared to wild-type AtNLP2, AtNLP2$^{M4}$ (W568L) significantly increased its complementation efficiency, while the others did not (Fig. 4d). AtNLP2$^{M4}$ not only induced the formation of more nodules, these nodules were also

markedly bigger. Sections of these nodules showed bacteria were released into nodule cells. Further, a meristem was formed at the apex, and vasculature in the peripheral tissue. This nodule phenotype is similar to that of the nodules formed on *Mtnin-1* roots trans-complemented by *PanNIN* (Fig. 4e–j and Supplementary Fig. 9).

To investigate whether the enhanced functioning of AtNLP2$^{M4}$ in nodulation is linked to increased nuclear localization, we compared

**Fig. 2 | The C-terminal DNA-binding domain of NIN orthologs from different lineages can function in nodule symbiosis. a** Electrophoretic mobility shift assay (EMSA) testing the binding of the C-terminus of different NIN orthologs to NIN binding sites of MtNF-YA1 and MtNF-YB16. Box plots show quantification of the bound fraction for each protein-probe combination. Three independent replicates were performed for NF-YA1 and four for NF-YB16. Asterisks indicate significant binding compared to empty vector control (Student's *t* test, 2-sided, *p* < 0.05). **b** qRT-PCR showing chimeric NIN proteins induce MtNF-YA1 and MtNF-YB16 expression in a transactivation assay. *Medicago* roots producing the N-terminus of MtNIN fused to the C-terminus of different NIN/NLPs and the rat glucocorticoid receptor were treated with dexamethasone (DEX) or DMSO (mock) for 16 h. Four independent biological replicates were used for the mock treatment and six for the dex treatment. Expression levels were normalized to the average expression of mock-treated empty vector roots. Asterisks indicate significant differences (Mann–Whitney *U*-test, two-sided, *p* < 0.05). **c** Complementation of Mtnin-1 mutants with chimeric proteins, consisting of the N-terminus of MtNIN, fused to the C-terminus of different NIN/NLPs. Plants were harvested at 4 weeks post inoculation with *Sinorhizobium meliloti* 2011 expressing GFP. Numbers below show the number of nodulating plants out of the total number of plants analyzed. Box plots show number of nodules per nodulated plant. Lowercase letters indicate significant differences between samples (Kruskal–Wallis and post-hoc Dunn's test, Benjamini–Yekutieli adjusted *p* < 0.05). Purple: symbiotic NIN, green: nonsymbiotic NIN. **d** Images of nodules formed on Mtnin-1 complemented with the chimeric proteins. Upper panels: stereomicroscope images; middle and bottom panels: longitudinal sections stained with toluidine blue. Numbers indicate nodules with released bacteria. The order of images from left to right reflects the phylogenetic distance to *Medicago*. Arrows: infection threads, arrowheads: released rhizobia. Scale bars: 2 mm (upper panels), 100 μm (middle panels) and 20 μm (bottom panels). Source data of **a**–**c** are provided as a Source data file. All box plots show the median (center line) and interquartile range (box). Points represent individual observations.

the subcellular localization of *GFP-AtNLP2*[M4] and *GFP-AtNLP2* in *Medicago* nodules. *GFP-AtNLP2*[M4] showed increased nuclear localization when compared to *GFP-AtNLP2* (Fig. 4k–m), suggesting that this modification likely enhances AtNLP2 function by promoting its nuclear localization.

Next, we tested whether introducing multiple mutations into AtNLP2 would further improve its function in nodule formation. Some combinations significantly improved complementation efficiency over wild-type AtNLP2, for example AtNLP2[M1,2,4,5], but not beyond AtNLP2[M4] (Fig. 4d and Supplementary Fig. 9a). Interestingly, AtNLP2[M1,2,5], without mutating position 4, also significantly enhanced nodule formation, although no rhizobial release was observed (Supplementary Fig. 10). These data demonstrate that multiple ways of amino acid modifications can improve NIN functioning in nodule symbiosis, providing alternative evolutionary trajectories enhancing symbiotic NIN efficiency.

We also tested whether the identified changes improved nodulation at zero nitrate growth conditions. This showed that none of the AtNLP2 mutant variants, including AtNLP2[M4], complemented *Mtnin-1* for nodulation (Supplementary Fig. 11). This demonstrates that a low concentration of exogenous nitrate is essential for the functioning of AtNLP2, AtNLP2[M4], and other AtNLP2 mutant variants, whereas this is not the case for symbiotic MtNIN and PanNIN.

Given the efficient functioning of AtNLP2[M4] in nodulation, we further investigated the evolutionary trajectory of position 4. It shows that the leucine residue at this position likely did not arise at the base of the NFC but instead represents the most probable ancestral state (Supplementary Fig. 7). Non-nodulating plant species, such as *Petunia axillaris* and *Vitis vinifera* both possess a NIN ortholog with a leucine at position 4. However, these proteins did not trigger nodule primordium formation when introduced into Mt*nin-1*, and only some infection threads were formed in the case of VvNIN (Supplementary Fig. 12). This shows that a leucine at position 4 alone is not sufficient to induce nodule formation.

Comparison of the region surrounding position 4 of NINs with paralogues showed that it is unique to the NIN subclade. Prior to the evolution of the NFC, both position 4 and its surrounding region were variable (Supplementary Fig. 7). In contrast, they became highly conserved in symbiotic NINs within NFC, highlighting strong evolutionary pressure to maintain this region and in line with its functional significance in nodulation.

## Discussion

In this study, we uncovered an intriguing aspect of the evolution of symbiotic NIN. Although its recruitment into symbiosis dramatically shifted its function from a nitrate-sensing to a symbiont-responsive transcription factor, nonsymbiotic NINs were already partially capable of performing this function. Nonsymbiotic NIN orthologs from a variety of species can induce symbiotic gene expression, enabling infection thread formation and the initiation of nodule organogenesis. However, they do this with low efficiency and without the ability to facilitate the intracellular colonization of nodule cells by rhizobia. Remarkably, a single amino acid change is sufficient to convert a nonsymbiotic NIN, AtNLP2, into a NIN protein functionally similar to the symbiotic PanNIN, including the ability to facilitate bacterial intracellular accommodation in nodules.

The properties of nonsymbiotic NINs in a nodulation context are variable with respect to their subcellular localization in response to nitrate, their ability to induce symbiotic gene expression, and their ability to induce rhizobium infection and nodule formation. This variability may result from relaxed selection pressure following the duplication of an NLP within eudicots, giving rise to the NIN and NLP1 clades[16]. Lineage-specific gene-duplication may have allowed further functional diversification as observed in *Arabidopsis* and cassava NIN orthologs, which differ in subcellular localization as well as in DNA-binding abilities. Symbiotic functionality arose within this context of naturally occurring variation across the NIN clade. It is retained without root nodule symbiosis as a selection pressure, and without a clear correlation between symbiotic functionality and evolutionary distance to the NFC.

In our study, AtNLP2 was the nonsymbiotic NIN ortholog with the best ability to function in nodule formation. This feature of AtNLP2 does not directly correlate with the functioning of its C-terminal DNA binding domain, for which, for example, tomato SlNIN performed better in inducing symbiotic target genes in *Medicago*. In line with this, we argue the symbiotic functioning of AtNLP2 most probably is related to its N-terminally regulated nuclear localization under low nitrate conditions, a prerequisite of a symbiotic NIN protein. The functionality of AtNLP2 in symbiosis could be further enhanced in multiple independent ways through introducing amino acid changes in its N-terminal region, which are under positive selection in the NFC, e.g., AtNLP2[M4] or AtNLP2[M1,2,5]. Besides these, there may still be additional residues involved in the adaptation of NIN for symbiosis, since mutating residues M1 to M7 in *Medicago* MtNIN still allows the formation of some pink nodules (Supplementary Fig. 8). Such additional residues essential for symbiotic NIN functioning might be, for example, under negative selection, or selected independently in different lineages of the NFC.

Amino acid changes, such as AtNLP2[M4] improve the symbiotic functionality of AtNLP2 to a level similar to that of PanNIN, enabling rhizobial release and symbiosome formation, but not full bacterial elongation, and leading to premature senescence. The lack of full complementation of the *Medicago nin* knockout mutant by both modified AtNLP2 and PanNIN reflects a NIN-controlled response evolved later in the legume lineage, after the initial recruitment of NIN in nodulation.

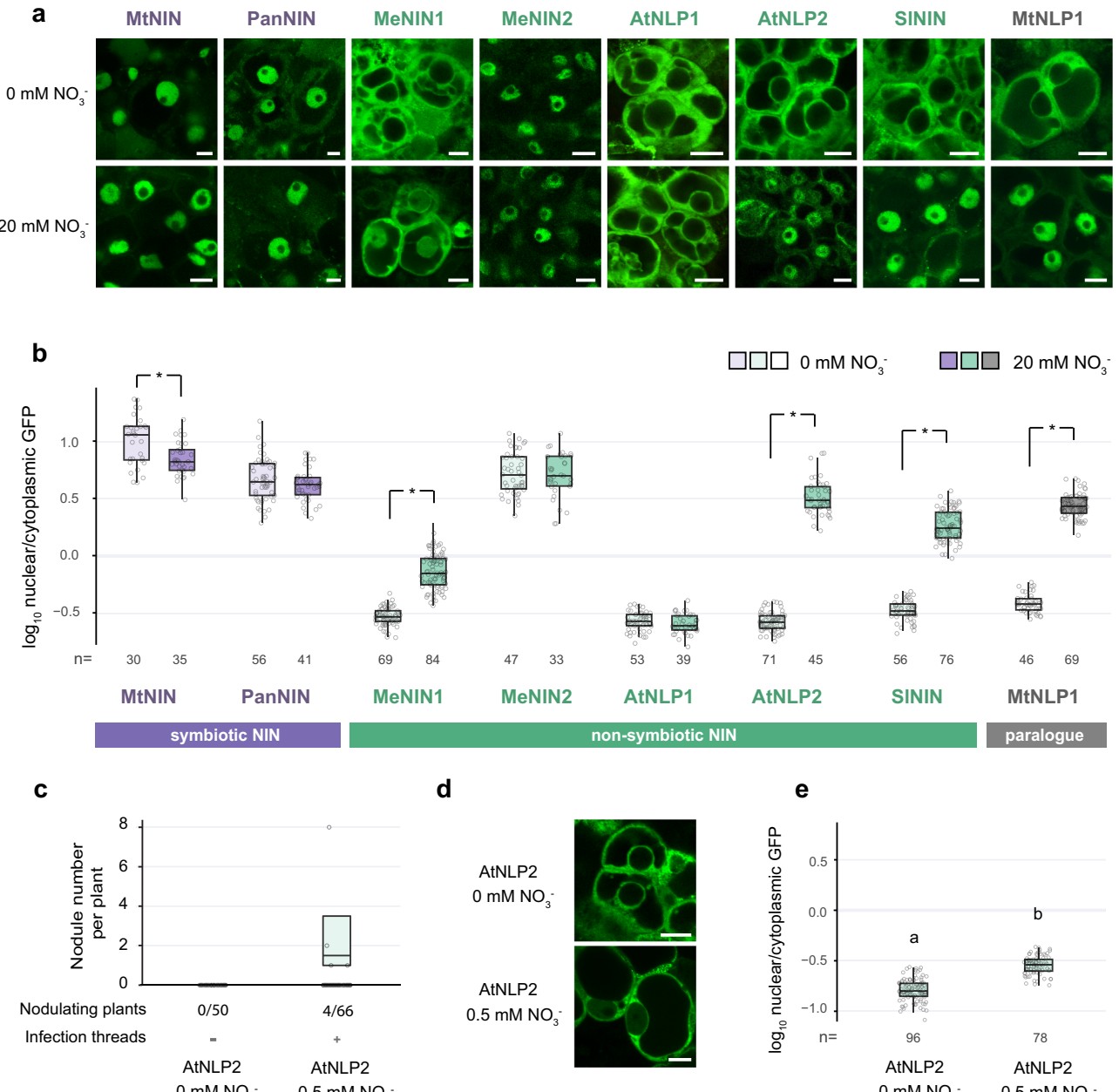

**Fig. 3 | Nuclear localization of NIN orthologs is required for symbiotic function.**
**a** Confocal image showing the subcellular localization of GFP-tagged NIN orthologs and a paralog in *Medicago* nodule infection zone cells at 0 mM and 20 mM nitrate. The order of images from left to right reflects the phylogenetic distance to *Medicago*. Scale bars: 10 μm. **b** Quantification of subcellular localization shown in (**a**); asterisks indicate significant differences (Mann–Whitney *U*-test, 2-sided, Benjamini–Yekutieli adjusted *p* < 0.05). **c** Number of nodules formed on Mtnin-1 mutant roots complemented with AtNLP2 at 0 mM nitrate or 0.5 mM nitrate. Numbers show the number of nodulating plants out of the total of plants analyzed.

Box plots show the number of nodules per nodulated plant. Differences are not significant (Mann–Whitney *U*-test, 2-sided, *p* < 0.05). **d** Confocal images showing subcellular localization of AtNLP2 under 0 mM or 0.5 mM nitrate. Scale bars: 10 μm. **e** Quantification of subcellular localization in (**d**). Lowercase letters indicate significant differences between samples (Mann–Whitney *U*-test, 2-sided, *p* < 0.05); *N* indicates the number of cells analyzed. Color code: purple: symbiotic NIN, green: nonsymbiotic NIN, gray: NIN paralog. Source data of (**b,e**) are provided as a Source data file. All box plots show the median (center line) and interquartile range (box). Points represent individual observations.

We used ancestral sequence reconstruction to filter out lineage-specific evolution and test the functionality of the predicted NIN proteins from the ancestor of the nitrogen-fixing clade ($NIN_{NFC}$), and its most recent nonsymbiotic predecessor ($NIN_{Rosids}$). As NIN contains variable regions, for which misprediction of a single amino acid can still disturb the protein function, the predicted ancestors likely underestimate the functionality of the true ancestors. Despite this limitation, the primordia induced by $NIN_{NFC}$ support that the common ancestor of the NFC possessed a NIN protein able to function in

nodulation. The functionality of predicted nonsymbiotic ancestor $NIN_{Rosids}$ in infection thread formation supports that this symbiotic property predates the evolution of nodulation.

Notably, the leucine introduced into AtNLP2$^{M4}$ (W568L) represents the most likely ancestral state. Although this residue is not a new invention underlying the evolution of symbiotic NIN, it is specifically conserved in NFC, consistent with its functional role in nodule symbiosis. This tryptophane to leucine substitution enhances the nuclear localization of AtNLP2$^{M4}$, possibly by causing a conformational shift

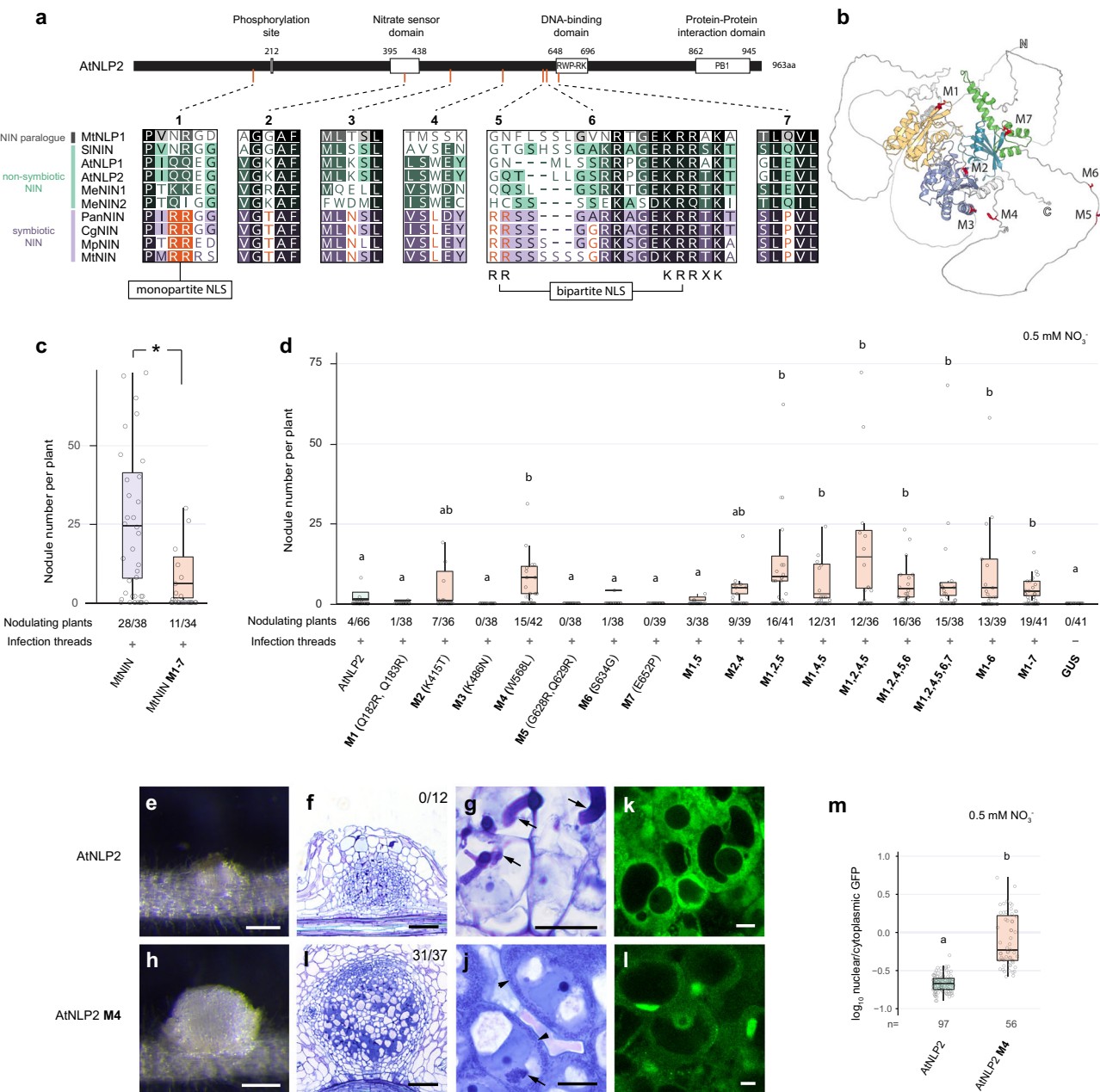

**Fig. 4 | A single amino acid adaptation improves AtNLP2 functioning in nodulation. a** Schematic illustration of seven conserved amino acid changes between symbiotic (purple) and nonsymbiotic (green) NIN orthologs. Amino acid residues in orange font or shade are conserved specifically in symbiotic NINs. Mt: *Medicago*, Sl: tomato, At: *Arabidopsis*, Me: cassava, Pan: *Parasponia*, Cg: *Casuarina glauca*, Mp: *Mimosa pudica*. **b** Structural prediction analysis of AtNLP2 forms three main domains; the N-terminal nitrate-responsive domain that forms two globes (yellow and purple), an RWP-RK DNA-binding domain (blue) and a PB1 protein–protein interaction domain (green). The interdomain regions are highly disordered (white). The positions of the seven conserved amino acids (M1–M7) are marked in red. **c** Number of nodules formed on Mtnin-1 mutant roots complemented with MtNIN and MtNIN^M1–7. Numbers show the number of nodulating plants out of the total number of plants analyzed. Box plots show the number of nodules per nodulated plant. Asterisk indicates significant difference (Mann–Whitney *U*-test, 2-sided, *p* < 0.05). **d** Number of nodules formed on Mtnin-1 mutant roots complemented with wildtype AtNLP2 and AtNLP2 with different adaptations at single or multiple positions. Numbers show the number of

nodulating plants out of the total number of plants analyzed. Lowercase letters indicate significant differences between samples (Kruskal–Wallis and post-hoc Dunn's test, Benjamini–Yekutieli adjusted *p* < 0.05). **e, h** Stereomicroscope images of nodules formed on Mtnin-1 complemented with AtNLP2 and AtNLP2^M4. Scale bars: 2 mm. **f, i** Longitudinal sections stained with toluidine blue. Numbers indicate the number of nodules with released bacteria per analyzed nodules. Scale bars: 100 μm. **g, j** Magnification of nodule cells. Arrows indicate infection threads; arrowheads indicate released rhizobia. Scale bars: 20 μm. **k, l** Confocal images showing the subcellular localization of wildtype AtNLP2 and AtNLP2^M4 in *Medicago* nodule infection zone cells under 0.5 mM nitrate. Scale bars: 10 mm.
**m** Quantification of subcellular localization shown in (**k, l**). Lowercase letters indicate significant differences between samples (Kruskal–Wallis and post-hoc Dunn's test, Benjamini–Yekutieli adjusted *p* < 0.05; *n* ≥ 50 nuclei from at least 7 nodules). Source data of (**c,d,m**) are provided as a Source data file. All box plots show the median (center line) and interquartile range (box). Points represent individual observations.

that either exposes the nuclear localization signal or masks the nuclear export signal. However, for AtNLP2$^{M4}$ to function in nodulation, some nitrate, even at low concentration, must be present. This indicates that a nitrate-dependent activation is still required. In AtNLP7, nitrate triggers a conformational change that derepresses the protein via its N-terminus[30], and a similar mechanism might be in place for AtNLP2. Due to current limitations in protein modeling, an experimentally determined structure of NLPs will be essential to provide insights into their nitrate-dependent structure and activity.

Taken together, our study provides insight into the evolutionary trajectory and molecular adaptations that allowed NIN to function as the central regulator of nitrogen-fixing nodule symbiosis. The findings support the hypothesis that ancestral NIN proteins possessed latent symbiotic capacity to control bacterial intracellular infection and nodule organogenesis. Following this recruitment, limited adaptations that reduce the dependency on nitrate are sufficient for NIN to evolve into an efficient regulator of nodulation.

## Methods

### Plant material and growth conditions

The *Medicago truncatula* (*Medicago*) *nin-1* knock-out mutant (in Jemalong A17 background), and wild-type Jemalong A17 plants were used in this study. *Agrobacterium rhizogenes* strain MSU440 was used for transformation, as described previously[37]. The composite plants were grown at 21 °C and 16 h light/8 h dark regime, in perlite saturated with Färhaeus medium, with Ca (NO$_3$)$_2$ added to the indicated NO$_3^-$ concentrations. For nodulation experiments, plants were first grown in perlite for 1 week, then inoculated with *Sinorhizobium meliloti* 2011 rhizobia constitutively expressing GFP (OD$_{600}$ = 0.1, 2 mL per plant), and harvested 4 weeks post-inoculation. Transgenic roots were selected for analysis based on expression of the visual selection marker mCherry. For the trans-activation assay, plants were grown for 3 weeks on perlite saturated with Färhaeus medium, without NO$_3^-$ added. Then, plants were treated with Färhaeus containing 10 μM dexamethasone (from a 10 mM stock in DMSO) or containing 0.1% DMSO as a mock treatment. The transgenic roots were harvested 16 h after treatment. For the subcellular localization studies, plants were grown in perlite saturated with Färhaeus medium, without NO$_3^-$. After growing in perlite for 1 week, the plants were inoculated with wildtype *S. meliloti* 2011 rhizobia (OD$_{600}$ = 0.1, 2 mL per plant). After 4 weeks post inoculation, the nodules were harvested for further analysis.

### Constructs

The constructs used in this study were assembled by golden gate cloning. Standard promoters, terminators, backbones, and binary vectors were obtained from the MoClo Toolkit and MoClo Plant Parts Kit[38,39], Addgene Kit #1000000044, and Kit #1000000047). Modules specific for this study were de novo synthesized. Supplementary Data 3 lists the sequences of all used golden gate cloning vectors.

To analyze the subcellular localization of different NINs, non-symbiotic NINs and MtNLP1 fused to GFP were driven by the constitutive *Lotus japonicus UBIQUITIN1* promoter (*pLjUBQ1*)[40]. When using this promoter to drive the expression of symbiotic *MtNIN* and *PanNIN* N-terminal *GFP* fusions, nodule formation was suppressed, likely due to autoregulation of nodulation controlled by symbiotic NIN[6]. Therefore, we used the nodule-specific *MtNIN* promoter to drive GFP-tagged *MtNIN* and *PanNIN*. Related constructs are included in Supplementary Data 3.

### Microscopy

The images of the nodules and the roots were taken using a stereo-microscope (M165 FC, Leica). For the section images, embedding of plant tissue in plastic, sectioning and tissue staining were performed as described previously[41]. Sections were analyzed using a DM5500B microscope equipped with a DFC425C camera (Leica). The subcellular

localization and infection threads were studied using a Leica SP8 confocal microscope.

To analyze the subcellular localization under different nitrate concentrations, nodules were hand-sectioned and treated with 0, 0.5, or 20 mM KNO$_3$ solution for 1 h before microscopy. The GFP intensity in the nucleus and cytoplasm was quantified using ImageJ software (1:53c) on the confocal pictures.

### Quantification of gene expression by qRT-PCR

RNA was isolated from *Medicago* roots using the EZNA Plant RNA kit (Omega Bio-Tek), following manufacturer's protocol. The iScript cDNA synthesis kit (Bio-Rad) was used to synthesize cDNA with a blend of oligo (dT) and random primers. Primers targeting symbiotic marker genes are listed in Supplementary Table 1. qRT-PCR was performed on the CFX96 connect Real-Time PCR system (Bio-Rad), using SYBR Green Supermix (Bio-Rad). The gene expression was normalized using *MtACTIN2* as a reference gene.

### Electromobility shift assay (EMSA)

Proteins for EMSA were produced using the TNT SP6 high-yield wheat germ expression system (Promega) using the vectors pL1M-R1 EMSA 6HA-MBP-MtNINct, 6HA-MBP-PanNINct, 6HA-MBP-MesNIN1ct, 6HA-MBP-MesNIN2ct, 6HA-MBP-SlNINct, 6HA-MBP-AtNLP2ct, 6HA-MBP-AtNLP1ct, 6HA-MBP-MtNLP1ct (Supplementary Data 3 and Supplementary Fig. 3c) as input. The in vitro translation was incubated for 2.5 h at 25 °C. Fluorescent probes were generated by PCR on probe template vectors pL0M-P NINBS MtCEP7, pL0M-P NINBS MtCLE13, pL0M-P NINBS MtLBD16, pL0M-P NINBS MtNF-YA1 or pL0M-P NINBS MtNF-YB16 (Supplementary Data 3) using primers conjugated to fluorescent dye IR700 (Supplementary Table 1). PCR fragments were purified using the GeneJet PCR purification kit (Thermo Scientific). Four microliters of the in vitro translation mix and 20 ng of the PCR probe were combined in EMSA binding buffer to a total volume of 12 μl (0.25 mg/ml BSA, 7.5 mM HEPES-NaOH pH 7.3, 0.7 μM DTT, 70 mM KCl, 1.5 mM MgCl$_2$, 60 μg/ml salmon sperm DNA, 2.5% CHAPS, 8% glycerol), and incubated for 30 min on ice. The reaction was run on a 0.5× TBE, 5% acryl-bisacrylamide gel, and visualized using a LiCor Odyssey fluorescence gel-scanner. The intensity of bands was analyzed using ImageJ software (1:53c).

### Western blot analysis

To visualize the synthesized protein used in EMSA, Mini-PROTEAN TGX Stain-Free Gels (Bio-Rad) were used for running the protein gels, then the samples were transferred to polyvinylidene difluoride membranes. After blocking with 3% bovine serum albumin, 5000 times diluted anti-HA-HRP (Miltenyi Biotec) with the ECL Western Blotting Substrate (Bio-Rad) were used for detection.

### Alignment and comparison of amino acid sequences

Protein sequences of NIN and NLP1 orthologs were identified by BLAST, and their orthology to MtNIN and MtNLP1 was confirmed based on their clustering with these proteins in a phylogenetic tree (Supplementary Fig. 7). Protein sequences and accession numbers are listed in Supplementary Data 4. A multiple sequence alignment was performed using the MAFFT alignment tool (1.2.2) in Geneious Prime software (GraphPad Software LLC). A phylogenetic tree was built using MrBayes (3.2.7a x86_64), using AtNLP7 as an outgroup, and without topology restraints.

To quantify conservation of amino acids in subgroups, the total alignment was subsetted to contain only the NFC or all dicot NIN orthologs. Jalview (2.11.4.1; The Barton Group, University of Dundee) was used to evaluate conservation levels of the subsets (Supplementary Data 2). The seven identified sites were selected based on their high conservation in NFC but low conservation across all dicot NIN orthologs, combined with manual evaluation.

## Ancestral sequence reconstruction

For ancestral sequence reconstruction, we used the tree built using MrBayes without topology constraints as a starting point to select the nodes for NIN reconstruction; the NFC clade ($NIN_{NFC}$), and the most recent nonsymbiotic ancestor, encompassing the NFC as well as species belonging to both Fabids and Malvids ($NIN_{Rosids}$). We used this tree and the alignment as input for GRASP (version 21-Mar-2024), using two independent methods of indel prediction; bi-directional edge parsimony (BEP) and simple indel-coding maximum likelihood (SICML). In parallel, we used MrBayes to reconstruct the ancestors, setting the nodes to be reconstructed as constraints. We compared the ancestors obtained by the three different methods and manually inspected and curated each position where the models disagree (Supplementary Data 2).

## Transient expression in *Nicotiana benthamiana* leaves

*Nicotiana benthamiana* (tobacco) were grown in soil for 4 weeks. The plants were treated with 20 mM $KNO_3$ or water (mock) 1 day before the infiltration. *Agrobacterium tumefaciens* strain C58 carrying constructs were co-infiltrated with silencing inhibitor P19 in tobacco leaves as described previously[42]. The infiltrated leaves were collected for confocal microscopy 3 days post infiltration.

## Statistical analysis

All statistical analyses were performed using R-studio software (Posit Software, PBC). Nodule number per plant, gene expression in trans-activation assay, and ratio of fluorescence in the nucleus and cytoplasm were not normally distributed (Shapiro–Wilk $p < 0.05$). Therefore, nonparametric tests were used in all analyses. For multiple comparisons, we used the Kruskall–Wallis test, with Dunn's test post hoc analysis. $p$-values were corrected for multiple testing using the Benjamini–Yekutieli method. For single or pairwise comparisons, the Mann–Witney $U$-test was used. Bound fraction in EMSA assays was normally distributed (Shapiro–Wilk $p > 0.05$), and were analzed using a Student's $t$ test versus control samples, with multiple testing correction using the Benjamini–Yekutieli method.

## NIN structure modeling

All models were predicted using Alphafold 3 with settings unaltered from the default, and with all shown models being representative of five simulations.

## Reporting summary

Further information on research design is available in the Nature Portfolio Reporting Summary linked to this article.

# Data availability

All data are available within this article and its Supplementary Information. Source data are provided with this paper.

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

## Acknowledgements

We thank Tom Peeters, Cloé Villard, Robin van Velzen, Joël Klein, Joan Wellink, Ella Köbben, and Stan van Wijk for their contribution and suggestions to this project. This project is supported by funding from the project Enabling Nutrient Symbioses in Agriculture (ENSA) that is funded by Bill and Melinda Gates Agricultural Innovations (INV-57461), China Scholarship Council (201506300062 to J.L., 201906170085 to S.Y., 202008150090 to M.L.), and the Dutch Science Organization (Nederlandse Organisatie voor Wetenschappelijk Onderzoek VI.Veni.212.132) to R.H.

## Author contributions

J.L. conceived and designed the work, performed the experiments, analyzed the data, and wrote the manuscript. S.Y. conceived and designed the work, performed the experiments, analyzed the data, and wrote the manuscript. M.L. performed experiments and analyzed the data. D.S. performed experiments. M.T. performed experiments. R.B. performed experiments and analyzed the data. K.R.A analyzed the data. F.V. performed experiments. O.K. analyzed the data. R.G. conceived and designed the work, analyzed the data and wrote the manuscript. T.B. conceived and designed the work, analyzed the data and wrote the manuscript. R.H. conceived and designed the work, performed the experiments, analyzed the data, and wrote the manuscript.

## Competing interests

The use of chimeric proteins and amino acid substitutions reported in this manuscript to increase symbiotic functionality of nonsymbiotic NIN orthologs is considered for patent application. 63/863,186 (New U.S. Provisional Application based on U.S. Provisional Application No. 63/677,618). Patent filed on 13-08-2025. Applicant: Wageningen Universiteit. Inventors; Jieyu Liu, Siqi Yan, Min Li, Rik Huisman, Ton Bisseling, and Rene Geurts. The other authors declare no competing interests.

## Additional information

**Supplementary information** The online version contains Supplementary material available at https://doi.org/10.1038/s41467-026-71330-1.

