## [Transparent Peer Review File · Nature Communications]

Ancestral Functionality and Symbiotic Refinement of NIN in Root Nodule Symbiosis

Corresponding Author: Dr Rik Huisman

Version 0:

Reviewer comments:

Reviewer #1

(Remarks to the Author)

In this manuscript, the authors investigate the evolutionary and functional divergence between symbiotic and non-symbiotic NIN/NLP orthologs. They first examined *Medicago truncatula* NIN (MtNIN) orthologs from non-symbiotic species to test whether these genes can complement and restore infection thread formation in *Medicago*. The authors found that *Manihot esculenta* NIN (MeNIN), *Arabidopsis* NLP2 (AtNLP2), and *Solanum lycopersicum* NIN (SiNIN) partially complemented infection thread formation. Next, they showed that the DNA-binding domains at the C-termini of non-symbiotic NIN orthologs can bind to canonical symbiotic NIN-binding sites, suggesting conservation of DNA-binding capability. To explore the contribution of different domains, they created chimeric proteins by fusing the N-terminus of MtNIN with the C-termini of non-symbiotic orthologs, and proposed that the N-terminus may control symbiotic specificity or function of NIN. The authors further analyzed seven amino acids conserved in symbiotic NINs and introduced single or combinatorial point mutations into AtNLP2. Interestingly, specific substitutions such as W586L (M4) or triple mutations (M1, M2, M5) enhanced AtNLP2's ability to promote nodulation, suggesting that certain conserved residues in NIN-like proteins contribute to symbiotic activity. Overall, the study presents intriguing phenotypic results and offers potential insights into the molecular evolution of NIN/NLP family proteins. However, several major concerns limit the impact and interpretability of the findings:

1. While the phenotypic observations are interesting, the manuscript lacks molecular or biochemical evidence explaining how these amino acid substitutions enhance nodulation. Without mechanistic data, the causal link between these mutations and enhanced symbiotic function remains speculative.

2. The rationale for selecting and mutating the seven conserved residues is not clearly explained. In particular, the M4 mutation (W586L) raises questions: the authors define a "conserved M4 fragment" based on MtNIN, yet no apparent sequence similarity exists between MtNIN and AtNLP2 at the aligned positions (Extended Data Fig. 7). It is unclear how the authors aligned the MtNIN L586 residue with the W residue in AtNLP2. A clear alignment justification or structural homology evidence is needed to support this comparison.

3. The conclusions regarding subcellular localization are not well supported. The authors rely solely on GFP fusion imaging without corroborating data. It remains unclear whether the reported "nucleolus" signals are indeed nucleolar or correspond to other subnuclear compartments. To strengthen these claims, the authors should perform additional validation such as co-localization with organelle markers, subcellular fractionation, or immunoblotting.

4. GFP-tagged proteins are prone to degradation or cleavage, which can result in misleading fluorescence patterns. The authors should verify that the observed fluorescence corresponds to intact fusion proteins rather than free GFP or truncated fragments, for example by immunoblotting using anti-GFP antibodies.

Reviewer #2

(Remarks to the Author)

The manuscript entitled “Evolution of a NIN-LIKE PROTEIN into a Master Regulator of Nitrogen-Fixing Nodulation” reports attempt to modify the protein sequence of non-legume NIN-Like proteins in order to increase their functionality to initiate nitrogen-fixing symbiotic nodulation, by complementing the Medicago truncatula model legume nin mutant.

- 1) The title, referring to symbiotic NIN as a “master regulator of nitrogen-fixing nodulation”, is somewhat unclear and overstated. Something like “Evolution of non-legume NIN-LIKE PROTEINS into a Symbiotic Regulator of Nitrogen-Fixing Nodulation” would be more appropriate.
- 2) l. 77, “while the other NIN orthologues of Arabidopsis and cassava” is unclear to me, do you mean NIN-LIKE proteins? Please clarify about which Arabidopsis/cassava NIN orthologues you are talking about here, maybe by using a phylogenetic tree?
- 3) l. 88, when mentioning “lineage-specific evolution still occurred”, this refers specifically to the late nodulation stages? Please clarify. Mention this interesting result at the end of the paragraph in the concluding sentence l. 102-104, and also in the discussion section. Currently, it is strange that this impact of NIN on late nodulation stages (premature senescence) is never discussed again in the whole manuscript.
- 4) l. 92, the concept of “ancestral sequence reconstruction to resurrect the NIN protein” should be better defined, either here, or even in the introduction, by explaining the molecular basis of this strategy. To me, “resurrect” seems strange here, please double check if appropriate (I am not a specialist of such evolutionary approaches). Similarly, the sentence l. 324 is difficult to understand, please help non-specialist readers.
- 5) l. 97, the use of “predisposition” is somewhat confusing as it has been in the last years overused in order to imply quite different concepts. Please try to use another wording, here and everywhere else in the manuscript.
- 6) l. 101, maybe give the frequency (%).
- 7) l. 104, replace “facilitating” by “being at the basis of”.
- 8) l. 106, figure 1 legend, “non-symbiotic” in the title is not appropriate, as MtNIN and PanNIN are included in this figure. The title could then be “NIN orthologs from different plants (lineages?) have a differential function in nodulation”. The same comment applies for the title of the Figure 2. The Figure 1 panels m and o should be moved at the end of the figure, to allow grouping together all stereomicroscopy images, as currently it cannot be easily understood why confocal images were shown only for two NIN proteins.
- 9) l. 124, replace by “occurred in THE NIN PROTEIN SEQUENCE”. In addition, “as they are transcription factor” sounds strange, please rephrase.
- 10) l. 141 and l. 186 (and maybe elsewhere in the manuscript), clarify that the symbiotic NIN proteins are MtNIN and PanNIN.
- 11) l. 143-145, “the induction OF A SUBSET OF PREVIOUSLY IDENTIFIED NIN targets”; “was variable DEPENDING ON TARGET GENES”; “some OF THESE symbiotic NIN targets”. Why the full-length proteins were not included in the main Figure 2? Provide a rationale or include.
- 12) l. 153, please speculate on why in the Figure 2c,d, AtNLP2 chimeras are not always very / the most effective. The same applies for non-chimeric proteins in the sup fig 5. When mentioning the different efficiency of complementation, l. 151-153, ideally classify them from the best to the worse complementation efficiency (or in the opposite order if you prefer). The same organization could similarly apply for ordering the images in the Figure 2d. Speculate in the discussion why in several cases MeNIN1 and SiNIN are similar or even perform better than AtNLP2, from left to right?
- 13) l. 159, “C-terminus SEQUENCE”.
- 14) In the Figure 2b, as well as in the sup Figure 5, the results of the statistical tests are sometime very unexpected. For example, in the sup Fig 5, some DEX inductions obtained with PanNIN (such as for MtNFYA-1 and MtCLE13 targets) seem clearly different from the control, even though not statistically significant. The same applies for some other transcription factor / target combinations, please double check. Conversely, some significant differences are not discussed, for example for MtNIN in the Fig 3b, even though I agree that this difference is likely non-biologically relevant.
- 15) l. 169, in the description of the panel c, clarify that chimeric proteins were used here.
- 16) l.180, add “(Figure2)” after “to induce nodules”.
- 17) l. 187, at least the reference Suzuki et al (2013) must be cited already here, and maybe a few others.
- 18) l. 189, explain how the efficiency of nuclear shuttling was quantified.
- 19) l. 190, add “subcellular” before “localization”. Speculate, here or in the discussion, about the surprising constitutive localizations of these two NIN / NLP proteins, either in the nucleus or in the cytoplasm.

- 20) l. 192, as said before, other NIN proteins than AtNLP2 partially complement the nin mutant, for some symbiotic responses at least. Would it be possible to improve the rationale of focusing only on AtNLP2, and not on these other NIN proteins? Alternatively, maybe MeNIN1 and SININ could be tested on the no nitrogen condition?
- 21) In the Fig 3b, the colors are very difficult to identify. Please indicate below the graph, with horizontal bars, what are the different categories, which I assume are the same as the ones shown in the Fig 1a, namely symbiotic NIN, non-symbiotic NIN, and NIN paralogs. Please do the same for the other figures where such color code is used, including in the Fig1 (by adding vertical bars on the right side of the tree) and Fig 4a (by adding vertical bars on the left side of the alignment). In this Figure 3b, there is no clear correlation with the previous nuclear localization data, please comment in the discussion about this.
- 22) l. 222, what is the consequence of this statement for nodulation evolution. Please clarify.
- 23) l. 224-231, nothing is mentioned about the position 3, is there indeed nothing relevant to highlight?
- 24) l. 235, in relation to the observation that AtNLP2 M1-7 is still able to form functional nodules, please indicate clearly in the discussion that this implies that there are other unknown residues that are still very relevant for explaining the function of NIN in nodulation. Similarly, l. 262, the requirement of nitrate for AtNLP2 M4 complementation efficiency indicate that other residues are involved, please also mention this clearly in the discussion.
- 25) l. 245, please comment also on the NIN M1,2,4,5 result, as it is never mentioned in the results. Does it have the same symbiotic efficiency as M4, or not, based on the Sup Fig 9b?
- 26) In the Fig 4, Sup Fig 9, and Sup Fig 10 images, some images are duplicated (AtNLP2 WT, M4, M1,2,4,5, and maybe others), which is not appropriate. Either do not include these images of the Fig 4 in the two supplementary figures, or alternatively, use independent biological replicate experiments.
- 27) l. 298-299, indicating that NLP is a nitrate sensor, and NIN a transcription factor, is confusing, as both proteins are transcription factors. Please improve to avoid misleading readers.
- 28) l. 305, do you want to mean "similar to PanNIN but not to MtNIN"? Please clarify.
- 29) l. 308, the sentences are not correct (this should be likely only one sentence).
- 30) l. 310, "some nitrate, EVEN AT LOW CONCENTRATION"
- 31) l. 312, "an experimentalLY"; l. 313, mentioning "the nitrate dependent activity" is unclear, please rephrase.
- 32) At the end of the discussion, in addition to the improvements already asked above, please discuss why do you think that you could not observe a full rescue of the NIN mutant with AtNLP2, and what would be the next steps as perspectives. Also discuss if mixing NIN promoter and protein sequence improvements at the same time would be a relevant perspective, or not.
- 33) Sup Fig 4, the letters indicating the RWP-RK and PB1 domains within the figure are too small, either increase their size, or use a color code (with clear-cut color differences).
- 34) The Sup Fig 7 is difficult to use, but I do not really see how to improve it. Please consider again if it is really useful and essential, or not.

Reviewer #3

(Remarks to the Author)

In this manuscript entitled "Evolution of a NIN-LIKE PROTEIN into a master regulator of Nitrogen-fixing Nodulation" Liu et al. studied the neofunctionalization of the transcription factor NIN for nodulation.

First, they tested the ability of NIN orthologs from diverse species to complement nodulation in the the Medicago nin mutant. They found that infection thread formation, and in one case organogenesis, can be recovered even with NIN from outside the nitrogen-fixing clade (NFC). The authors used this knowledge to compare sequences and identify that the C-terminal DNA-binding domain is not discriminating between complementing and non-complementing NIN orthologs. By contrast, they found that the increased nuclear localization of NIN was important for its symbiotic function, and linked with putative NLS and several mutations. Studying the evolution of these specific residues, the authors found that their gain predates the evolution of nodulation. Altogether, this suggests that an ancestral NIN, predating nodulation, was already fully functional for its recruitment in a symbiotic context, which is the main conclusion of the manuscript.

The study is based on very solid data: the complementation assays are conducted in the best possible way and scoring all relevant phenotypes (ITs, organogenesis, bacterial release...), the nuclear-localization assays are done in situ, and the EMSA include the appropriate controls. There is a lot of work presented here.

My only major comment is regarding the title. I find it not aligned with the conclusion. "Evolution.. into.." suggests a transition from an ancestral, non-symbiotic, state to a symbiotic one at the onset of nodulation. As the authors show, NIN was actually already symbiotically-functional before nodulation. The title should be rephrased in a way that reflects the conclusion. Understanding how NIN became integrated into its nodulation-related function will require further work, but this study significantly advance the field in that key direction!

Version 1:

Reviewer comments:

Reviewer #1

(Remarks to the Author)

The explanations are acceptable, and the issue has been strengthened in the revised manuscript.

Reviewer #2

(Remarks to the Author)

The revised manuscript now entitled "Ancestral functionality and symbiotic refinement of NIN in root nodule symbiosis" was greatly improved and answered all my concerns. A few minor corrections are still needed before publication.

1) About the title: I found the new title not catchy enough regarding the interest of the study. Maybe something like "Molecular improvement of a non-legume NIN-LIKE PROTEIN into a Symbiotic Regulator of Nitrogen-Fixing Nodulation" could be more clear and attractive?

2) In the abstract l. 28, "making it functionally comparable to a symbiotic NIN" is too vague and may be misleading for readers, for example by implying that it is as efficient as a complementation of the nin mutant by MtNIN. So please, be more specific. Maybe this could be done just by adding "to the Parasponia PanNIN symbiotic NIN" in the sentence?

3) The Figure 1a seems still to small to me, please increase the size of the letters?

4) L. 150, "attached to the GR" sounds strange to me, please consider to modify.

5) In the Figure 2b, it is unfortunate that MeNIN2 does not lead to a significant difference on the NF-YA1 regulation, as this does not fit with the text l. 154. I know that in the Extended Data Figure 4 all other NIN target genes are significantly upregulated by MeNIN2, but this is somewhat confusing, so please consider to show in the main Figure 2b another target than NF-YA1 that would exactly fit the what is said in the sentence l. 154.

6) l. 162, replace "constructs" by "chimeric proteins" (this was one of my previous requests already, which may have been not clear enough to you though).

7) In the Figure 2d legend, please indicate that "the order of images from left to right reflects the phylogenic distance to Medicago". This could (should?) be also mentioned in the other figures where this ordering is used. On a related line, mention in the text of the results l. 170 that "the degree of complementation does not correlate with evolutionary distance", which to me seems an interesting observation. L. 171, the ",", should be removed.

8) Following my request, you changed the statistical test used in the Extended Data Figure 5, but the legend still reads "Student's t test"! Please correct, and double check all figure legends in order to correct potential similar mistakes about the statistical tests cited.

9) Thanks for the extensive modification of the discussion that is now very nice!

Reviewer #3

(Remarks to the Author)

My main concern was regarding the title. The edited version reflects the conclusion and the data presented in the manuscript.

I do not have additional comments.

REVIEWER COMMENTS

Reviewer #1 (Remarks to the Author):

In this manuscript, the authors investigate the evolutionary and functional divergence between symbiotic and non-symbiotic NIN/NLP orthologs. They first examined *Medicago truncatula* NIN (MtNIN) orthologs from non-symbiotic species to test whether these genes can complement and restore infection thread formation in *Medicago*. The authors found that *Manihot esculenta* NIN (MeNIN), *Arabidopsis* NLP2 (AtNLP2), and *Solanum lycopersicum* NIN (SiNIN) partially complemented infection thread formation. Next, they showed that the DNA-binding domains at the C-termini of non-symbiotic NIN orthologs can bind to canonical symbiotic NIN-binding sites, suggesting conservation of DNA-binding capability. To explore the contribution of different domains, they created chimeric proteins by fusing the N-terminus of MtNIN with the C-termini of non-symbiotic orthologs, and proposed that the N-terminus may control symbiotic specificity or function of NIN. The authors further analyzed seven amino acids conserved in symbiotic NINs and introduced single or combinatorial point mutations into AtNLP2. Interestingly, specific substitutions such as W586L (M4) or triple mutations (M1, M2, M5) enhanced AtNLP2's ability to promote nodulation, suggesting that certain conserved residues in NIN-like proteins contribute to symbiotic activity. Overall, the study presents intriguing phenotypic results and offers potential insights into the molecular evolution of NIN/NLP family proteins. However, several major concerns limit the impact and interpretability of the findings:

1. While the phenotypic observations are interesting, the manuscript lacks molecular or biochemical evidence explaining how these amino acid substitutions enhance nodulation. Without mechanistic data, the causal

link between these mutations and enhanced symbiotic function remains speculative.

> Thank you for your positive feedback and for evaluating our manuscript. However, we respectfully disagree with the assertion that we do not provide a causal link between the identified mutations and the enhanced symbiotic function of the mutant variants of the non-symbiotic NIN protein AtNLP2.

(i) We identify specific amino acid substitutions, such as M4 and M1,2,5, that convert AtNLP2 into a protein that is functionally similar to symbiotic NIN (Figure 4d, Extended Data Fig. 9). This establishes a causal link between defined amino acid adaptations and enhanced symbiotic functionality.

(ii) We demonstrate that AtNLP2-M4 exhibits increased nuclear localization at low nitrate levels compared to wild-type AtNLP2 (Figure 4k-m), which is the most likely explanation for its enhanced functionality.

Further elucidating the molecular mechanism by which these substitutions improve the nitrate-independent nuclear localization of AtNLP2 would require a crystal structure of NIN and AtNLP2. We consider such studies timely and highly relevant, and is now mentioned in the Discussion section of the manuscript (lines 381-383). However, such an approach lies beyond the scope of the present work.

2.

The rationale for selecting and mutating the seven conserved residues is not clearly explained. In particular, the M4 mutation (W586L) raises questions: the authors define a “conserved M4 fragment” based on MtNIN, yet no apparent sequence similarity exists between MtNIN and AtNLP2 at the aligned positions (Extended Data Fig. 7). It is unclear how the authors aligned the MtNIN L586 residue with the W residue in AtNLP2. A clear

alignment justification or structural homology evidence is needed to support this comparison.

> The selection of the seven residues was based on the observation that these amino acids are specifically conserved in symbiotic NIN proteins when compared with non-symbiotic NIN proteins, suggesting positive selection in the NFC. This approach is summarised in Supplementary Table 2. We revised Extended Data Fig. 7 to better highlight the relevant regions. In the updated figure, we now explicitly label “symbiotic NIN,” “non-symbiotic NIN,” and “NIN paralogs” alongside the phylogenetic tree, and we emphasize MtNIN and AtNLP2 in bold to guide the readers.

In the specific case of M4, the revised figure more clearly shows that this region is conserved and well aligned among NIN orthologs (green shading for non-symbiotic NIN orthologs and purple shading for symbiotic NIN orthologs), but is less conserved among NIN paralogues (gray shading), including the outgroups AtNLP7, monocot NLPs, and the NLP1 orthogroup. Within the NIN orthologs, non-symbiotic NIN proteins (green shading) predominantly encode tryptophan (W) at position M4, whereas symbiotic NIN proteins consistently encode leucine (L) at this position. This NFC-specific conservation motivated the selection of this residue for further investigation.

3.

The conclusions regarding subcellular localization are not well supported. The authors rely solely on GFP fusion imaging without corroborating data. It remains unclear whether the reported “nucleolus” signals are indeed nucleolar or correspond to other subnuclear compartments. To strengthen these claims, the authors should perform additional validation such as co-localization with organelle markers, subcellular fractionation, or immunoblotting.

> We used confocal microscopy-based subcellular localization studies exclusively to distinguish cytoplasmic from nuclear localization, both of

which can be readily differentiated. This strategy is also consistent with previous NLP localization studies (Lin et al., 2018, doi:10.1038/s41477-018-0261-3; Nishida et al., 2018; doi:10.1038/s41467-018-02831, Misawa et al., 2022; doi: 10.1093/plcell/koac046, Durand et al., 2023, doi:10.1093/plcell/koad025). For example, the same method was used to show that *Arabidopsis* AtNLP7, the most well-studied NIN-like protein, exhibits cytoplasmic localization under low-nitrate conditions and accumulates in the nucleus under high-nitrate conditions (Liu et al., 2017, doi:10.1038/nature22077).

We did not draw conclusions about localization within specific subnuclear compartments, such as the nucleolus, and the term “nucleolus” is not mentioned anywhere in the manuscript. Therefore, additional studies involving co-localization with organelle markers, subcellular fractionation, or immunoblotting fall outside the scope of our research aims.

4.

GFP-tagged proteins are prone to degradation or cleavage, which can result in misleading fluorescence patterns. The authors should verify that the observed fluorescence corresponds to intact fusion proteins rather than free GFP or truncated fragments, for example by immunoblotting using anti-GFP antibodies.

> In our study, we focused on nitrate-dependent subcellular localization. We observed nitrate-dependent changes of multiple non-symbiotic NINs in the same samples under different nitrate conditions, as well as constitutive nuclear localization of symbiotic NINs. Such patterns cannot be attributed to free GFP, which typically localizes uniformly to both the cytoplasm and the nucleus (Haselhoff et al., 1997; doi.org/10.1073/pnas.94.6.2122).

We have clarified our method in the manuscript:

L. 201-206 *"For this, NIN orthologs and paralog MtNLP1 fused to GFP were expressed in a wildtype background. Rhizobium inoculated Medicago plants were grown in absence of nitrate for 4 weeks. Nodules of these plants were used for subcellular localization studies; first in absence of nitrate and subsequently after 1 hour of 20 mM KNO₃ treatment (Fig 3a). We quantified the ratio of GFP intensity in nucleus and cytoplasm under both conditions to visualize nitrate-dependent subcellular localization (Fig. 3b)."*

As mentioned in our answer to question 3, this method is widely used, and our observations are consistent with earlier reports. Although immunoblots of plant extracts containing GFP-tagged proteins often display bands corresponding to free GFP, this does not imply that the signals observed in microscopy arise from free GFP or truncated fragments of the chimeric proteins.

Reviewer #2 (Remarks to the Author):

The manuscript entitled "Evolution of a NIN-LIKE PROTEIN into a Master Regulator of Nitrogen-Fixing Nodulation" reports attempt to modify the protein sequence of non-legume NIN-Like proteins in order to increase their functionality to initiate nitrogen-fixing symbiotic nodulation, by complementing the Medicago truncatula model legume nin mutant.

> Thank you for reviewing our manuscript with such attention to detail and for providing clear suggestions for improvement. As outlined below, we accepted most of your suggestions, with a few exceptions that we discuss in more detail.

1) The title, referring to symbiotic NIN as a "master regulator of nitrogen-fixing nodulation", is somewhat unclear and overstated. Something like "Evolution of non-legume NIN-LIKE PROTEINS into a Symbiotic Regulator of Nitrogen-Fixing Nodulation" would be more appropriate.

> We changed the title accordingly. It reads now:

Ancestral Functionality and Symbiotic Refinement of NIN in Root Nodule Symbiosis.

2) l. 77, "while the other NIN orthologues of Arabidopsis and cassava" is unclear to me, do you mean NIN-LIKE proteins? Please clarify about which Arabidopsis/cassava NIN orthologues you are talking about here, maybe by using a phylogenetic tree?

> We have changed this sentence to be more specific (Line 77-78); 'other NIN orthologues' is changed into: 'AtNLP1, AtNLP3 and MeNIN2'.

We have marked symbiotic NIN and non-symbiotic NIN more clearly in the phylogenetic tree in Fig. 1a.

3) l. 88, when mentioning "lineage-specific evolution still occurred", this refers specifically to the late nodulation stages? Please clarify. Mention this interesting result at the end of the paragraph in the concluding sentence l. 102-104, and also in the discussion section. Currently, it is strange that this impact of NIN on late nodulation stages (premature senescence) is never discussed again in the whole manuscript.

> Indeed, we refer to the late stages of nodulation; differentiation and maintenance of bacteroids. We have now included lines in this paragraph (Line 87-90) and the discussion (line 360-363) to discuss this result.

Line 87-90: *'The differentiation and maintenance of bacteroids in Medicago nodules after release could depend on adaptations in NIN that are not present in PanNIN. As in Parasponia nodules, rhizobium bacteria are not released and elongated, PanNIN would not require these adaptations'*.

Line 360-63: *'similar to that of PanNIN, enabling rhizobial release and symbiosome formation, but not full bacterial elongation, and leading to premature senescence. The lack of full complementation of the Medicago*

nin knockout mutant by both modified AtNLP2 and PanNIN reflects a NIN-controlled response evolved later in the legume lineage, after the initial recruitment of NIN in nodulation’.

4) l. 92, the concept of “ancestral sequence reconstruction to resurrect the NIN protein” should be better defined, either here, or even in the introduction, by explaining the molecular basis of this strategy. To me, “resurrect” seems strange here, please double check if appropriate (I am not a specialist of such evolutionary approaches). Similarly, the sentence l. 324 is difficult to understand, please help non-specialist readers.

> We have clarified the concept. The passage now reads:

Line 95-104. *‘To further investigate the evolution of NIN, we used ancestral sequence reconstruction, a technique to infer the most probable sequence of internal nodes of a phylogenetic tree of a protein family. These sequences are an approximation of the ancestral states of the protein family during evolution^{23,24}. For this we analysed the protein sequence of a wide range of dicot NIN orthologs, paralogs from the NIN sister clade containing MtNLP1, as well as monocot NIN/NLP1 homologs that predate the split of the NIN and NLP1 orthogroups (Supplemental table 5). We used two independent ancestral sequence reconstruction algorithms, MrBayes²⁵ and GRASP²⁴, followed by manual curation. With this approach, we predicted the NIN protein sequences of the most recent ancestor of the nitrogen fixing clade (NINNFC) and the most recent non-symbiotic ancestor of the Rosid clade (NINRosids) (Extended Data Fig. 7, Supplementary Table 2).’*

The word resurrection of ancestral sequences is more often used to express and characterize predicted proteins, but since it is not essential, we removed it. The text now reads:

Line 107-108: *“We used gene synthesis to generate Medicago codon optimized DNA sequences, to functionally characterize the predicted ancestral proteins.”*

5) l. 97, the use of "predisposition" is somewhat confusing as it has been in the last years overused in order to imply quite different concepts. Please try to use another wording, here and everywhere else in the manuscript.

> We replaced 'predisposition' for 'latent capacity'. The sentence reads now: '.....supporting a latent capacity for symbiotic functioning of the ancestral Rosid NIN' (Line 109-110).

6) l. 101, maybe give the frequency (%).

> The nodule primordia that we observed are limited to a few dividing cell layers. This requires sectioning of the root to accurately identify. Therefore, it is not feasible to quantify the frequency of primordium formation.

7) l. 104, replace "facilitating" by "being at the basis of".

> We have changed the text accordingly.

8) l. 106, figure 1 legend, "non-symbiotic" in the title is not appropriate, as MtNIN and PanNIN are included in this figure. The title could then be "NIN orthologs from different plants (lineages?) have a differential function in nodulation". The same comment applies for the title of the Figure 2.

> We have adapted the titles. They now read;

Figure 1: '*NIN orthologs from different lineages are partially functional in nodule symbiosis*'

Figure 2: '*The C-terminal DNA binding domain of NIN orthologs from different lineages can function in nodule symbiosis*'

The Figure 1 panels m and o should be moved at the end of the figure, to allow grouping together all stereomicroscopy images, as currently it cannot be easily understood why confocal images were shown only for two NIN proteins.

> We have changed the figure order accordingly.

9) l. 124, replace by "occurred in THE NIN PROTEIN SEQUENCE". In addition, "as they are transcription factor" sounds strange, please rephrase.

> We have changed the text accordingly.

10) l. 141 and l. 186 (and maybe elsewhere in the manuscript), clarify that the symbiotic NIN proteins are MtNIN and PanNIN.

> We specified this as suggested.

11) l. 143-145, "the induction OF A SUBSET OF PREVIOUSLY IDENTIFIED NIN targets"; "was variable DEPENDING ON TARGET GENES"; "some OF THESE symbiotic NIN targets".

> We changed the text, it now reads:

Line 155-158: 'The induction of the tested NIN targets by the symbiotic NIN-chimers was generally strong, while the induction by non-symbiotic NIN-chimers was variable depending on the target gene. Most non-symbiotic NIN orthologs could induce some of these symbiotic NIN targets, but none of them significantly induced all targets.'

Why the full-length proteins were not included in the main Figure 2?
Provide a rationale or include.

> We did not include the full-length proteins as a main figure because we aim to study the functionality of the C-terminal DNA-binding properties of the various NIN orthologs. For this purpose, the use of chimeric proteins is more appropriate. The N-terminal half differs between symbiotic and non-

symbiotic NINs. The use of full-length NIN-GR fusions is therefore 'less clean', as the nuclear localization of non-symbiotic NIN proteins requires the presence of nitrate in the plant growth. Such growth conditions are likely to activate other Medicago NLP proteins (homologs of NIN), which could interfere with NIN gene expression resulting in an increased variation in NIN responses.

Therefore, we include the chimeric NINs in the main Figure, and use the full-length proteins in extended Data to support the conclusion.

12) l. 153, please speculate on why in the Figure 2c,d, AtNLP2 chimers are not always very / the most effective. The same applies for non-chimeric proteins in the sup fig 5.

> We adapted the text (l. 170-174):

'The different levels of symbiotic functionality of the full length proteins (Fig. 1), does not always correlate with their DNA binding efficiency and transactivation of symbiotic NIN-targets (Fig. 2). These data imply that the functional difference of NIN orthologs in nodulation, including the release and differentiation of rhizobia, is mainly associated with their N-terminal sequence.'

When mentioning the different efficiency of complementation, l. 151-153, ideally classify them from the best to the worse complementation efficiency (or in the opposite order if you prefer).

In the text, the phenotypes of the chimeric non-symbiotic NIN constructs are described from worst to best.

Line 164-166: *'On these roots, nodules were formed with phenotypes ranging from infected primordia (MeNIN2), release of rhizobia in nodule cells (MeNIN1), disorganized but differentiated rhizobia (AtNLP2), to radially organized differentiated rhizobia in pink nodules (SININ) (Fig. 2d).'*

The same organization could similarly apply for ordering the images in the Figure 2d.

In the figure, we prefer to maintain the chosen order because it reflects the phylogenetic distance to *Medicago*, which is used consistently across all figures (*Medicago*, *Parasponia*, cassava, *Arabidopsis*, and tomato). In our view, this avoids confusion and underscores an important message: the degree of complementation does not correlate with evolutionary distance.

Speculate in the discussion why in several cases MeNIN1 and SININ are similar or even perform better than AtNLP2, from left to right?

The ability to activate symbiotic NIN target genes falls within the naturally occurring variation across non-symbiotic NIN orthologs and was likely present before NIN was recruited into root nodule symbiosis. Moreover, the enhanced ability of AtNLP2 to function in nodule formation is apparently not due to a greater capacity to activate NIN target genes, as now discussed at the end of this paragraph (as above, line 170-174). This has let us focus on the properties of the N-terminal half of the protein.

We added to the discussion:

Line 346- 351: 'In our study, AtNLP2 was the non-symbiotic NIN ortholog with the best ability to function in nodule formation. This feature of AtNLP2 does not directly correlate with the functioning of its C-terminal DNA binding domain, for which for example tomato SININ performed better in inducing symbiotic target genes in Medicago. In line with this, we argue the symbiotic functioning of AtNLP2 most probably is related to its N-terminally regulated nuclear localization under low nitrate conditions, a prerequisite of a symbiotic NIN protein.'

13) l. 159, "C-terminus SEQUENCE".

> We have changed the text accordingly.

14) In the Figure 2b, as well as in the sup Figure 5, the results of the statistical tests are sometime very unexpected. For example, in the sup

Fig 5, some DEX inductions obtained with PanNIN (such as for MtNFYA-1 and MtCLE13 targets) seem clearly different from the control, even though not statistically significant. The same applies for some other transcription factor / target combinations, please double check. Conversely, some significant differences are not discussed, for example for MtNIN in the Fig 3b, even though I agree that this difference is likely non-biologically relevant.

> Thank you for pointing this out.

For Extended Data Figure 5, We re-considered the statistics, and realized that we applied a t-test on non-normally distributed data, resulting in non-significant outcomes that seem unexpected. We now log-transformed the data, resulting in normally distributed data (Shapiro-Wilk test), justifying the use of a t-test. Now, some more of the comparisons are significant. Our conclusions remain unaffected.

For Fig. 2b, we use Mann-Whitney *U*-test which is proper for this data, as they are not normally distributed, irrespective of log-transformation.

15) l. 169, in the description of the panel c, clarify that chimeric proteins were used here.

> The reviewer might have overlooked a phrase of the caption of Figure 2c. It states: "*Complementation of Mtnin-1 mutants with chimeric proteins, consisting of the N-terminus of MtNIN, fused to the C-terminus of different NIN/NLPs*".

16) l.180, add "(Figure2)" after "to induce nodules".

> We have changed the text accordingly.

17) l. 187, at least the reference Suzuki et al (2013) must be cited already here, and maybe a few others

> We now cite Suzuki et al (2013) accordingly.

18) l. 189, explain how the efficiency of nuclear shuttling was quantified.

> We have expanded this section to better explain the experiment. It now reads:

Line 201-206: 'For this, NIN orthologs and paralog MtNLP1 fused to GFP were expressed in a wildtype background. Rhizobium inoculated Medicago plants were grown in absence of nitrate for 4 weeks. Nodules of these plants were used for subcellular localization studies; first in absence of nitrate and subsequently after 1 hour of 20 mM KNO₃ treatment (Fig 3a). We quantified the ratio of GFP intensity in nucleus and cytoplasm under both conditions to visualize nitrate-dependent subcellular localization (Fig 3b).'

19) l. 190, add "subcellular" before "localization". Speculate, here or in the discussion, about the surprising constitutive localizations of these two NIN / NLP proteins, either in the nucleus or in the cytoplasm.

> We adapted the text as suggested, and added some discussion about their surprising localization.

L210-214: 'The subcellular localization of AtNLP1 and MeNIN2 was nitrate independent; AtNLP1 remained in the cytoplasm, whereas MeNIN2 was nuclear localized in both conditions (Fig. 3a,b). MeNIN2 might function similarly as what has been reported for AtNLP8, which is constitutively nuclear localized but requires nitrate to activate downstream gene expression.'

20) l. 192, as said before, other NIN proteins than AtNLP2 partially complement the nin mutant, for some symbiotic responses at least. Would it be possible to improve the rationale of focusing only on AtNLP2, and not

on these other NIN proteins? Alternatively, maybe MeNIN1 and SININ could be tested on the no nitrogen condition?

> We focused our detailed analyses on AtNLP2 because it is the most effective non-symbiotic NIN in rescuing nodule symbiosis, providing the strongest contrast under different nitrate conditions.

MeNIN1 has also been tested under no nitrogen condition, no infection threads were observed. This data now is included (line 224-225).

21) In the Fig 3b, the colors are very difficult to identify. Please indicate below the graph, with horizontal bars, what are the different categories, which I assume are the same as the ones shown in the Fig 1a, namely symbiotic NIN, non-symbiotic NIN, and NIN paralogs. Please do the same for the other figures where such color code is used, including in the Fig1 (by adding vertical bars on the right side of the tree) and Fig 4a (by adding vertical bars on the left side of the alignment).

> We adapted the Figures accordingly.

In this Figure 3b, there is no clear correlation with the previous nuclear localization data, please comment in the discussion about this.

> Figure 3b is based on the quantification of the green fluorescent signal as captured in confocal images of nodule cells, of which representative images are shown in Figure 3a.

In addition, the subcellular localization of AtNLP2, MtNLP1 and MtNIN are also consistent with previous studies (Line 209). (Durand et al., 2023, doi.org/10.1093/plcell/koad025; Lin et al., 2018, doi.org/10.1038/s41477-018-0261-3; Liu et al., 2021, [doi/10.1111/nph.17215](https://doi.org/10.1111/nph.17215)).

22) l. 222, what is the consequence of this statement for nodulation evolution. Please clarify.

> The statement that this phosphorylation site has been lost multiple times after the recruitment of NIN, means that the loss of this site was not required for NIN to be recruited in nodulation. We have added this line to clarify.

Lines 249-250: *'Thus, loss of this phosphorylation site was not a prerequisite for NIN to be recruited in root nodule symbiosis'*.

23) l. 224-231, nothing is mentioned about the position 3, is there indeed nothing relevant to highlight?

> Position 3 does not show a clear association with nuclear localization or known functional domains, and this mutation has no improved complementation efficiency when compared to AtNLP2 wt. We therefore don't think it needs additional highlighting.

24) l. 235, in relation to the observation that AtNLP2 M1-7 (MtNIN M1-7) is still able to form functional nodules, please indicate clearly in the discussion that this implies that there are other unknown residues that are still very relevant for explaining the function of NIN in nodulation.

Similarly, l. 262, the requirement of nitrate for AtNLP2 M4 complementation efficiency indicate that other residues are involved, please also mention this clearly in the discussion.

> This has now been mentioned in the discussion:

L. 354-358: *'Besides these, there may still be additional residues involved in the adaptation of NIN for symbiosis, since mutating residues M1 to M7 in Medicago MtNIN still allows the formation of some pink nodules (extended data fig 8). Such additional residues essential for symbiotic NIN functioning might be for example under negative selection, or selected independently in different lineages of the NFC.'*

25) I. 245, please comment also on the NIN M1,2,4,5 result, as it is never mentioned in the results. Does it have the same symbiotic efficiency as M4, or not, based on the Sup Fig 9b?

> Indeed, some combinations have a similar symbiotic efficiency as AtNLP2-M4, based on the statistics shown in Fig. 4d and Extended Data Fig. 9b. We mention this now in the text.

Line 280-282: '*Some combinations significantly improved complementation efficiency over wild-type AtNLP2, for example AtNLP2M1,2,4,5, but not beyond AtNLP2M4 (Fig. 4d and Extended Data Fig. 9a).*'

26) In the Fig 4, Sup Fig 9, and Sup Fig 10 images, some images are duplicated (AtNLP2 WT, M4, M1,2,4,5, and maybe others), which is not appropriate. Either do not include these images of the Fig 4 in the two supplementary figures, or alternatively, use independent biological replicate experiments.

> Extended Data Figures 9 and 10 are supplementary to the main Figure 4d and were derived from the same experiment. The same representative images were initially used to facilitate direct comparison across mutations, which we considered appropriate. To address the reviewer's concern, we have now replaced these panels with images from independent biological replicate.

27) I. 298-299, indicating that NLP is a nitrate sensor, and NIN a transcription factor, is confusing, as both proteins are transcription factors. Please improve to avoid misleading readers.

> We adapted the text aiming to avoid this confusion. We now refer to NLP as a 'nitrate sensing transcription factor'.

28) l. 305, do you want to mean "similar to PanNIN but not to MtNIN"?
Please clarify.

Line 359-360:

'Amino acid changes such as AtNLP2M4 improve the symbiotic functionality of AtNLP2 to a level similar to that of PanNIN'

29) l. 308, the sentences are not correct (this should be likely only one sentence).

> We rewrote the sentences.

30) l. 310, "some nitrate, EVEN AT LOW CONCENTRATION"

> We adapted the text as suggested.

31) l. 312, "an experimentally"; l. 313, mentioning "the nitrate dependent activity" is unclear, please rephrase.

> We adapted the text as suggested.

32) At the end of the discussion, in addition to the improvements already asked above, please discuss why do you think that you could not observe a full rescue of the NIN mutant with AtNLP2, and what would be the next steps as perspectives. Also discuss if mixing NIN promoter and protein sequence improvements at the same time would be a relevant perspective, or not.

> As described in our discussion, we believe that the maximum level of complementation that can be achieved is that of PanNIN; and the next steps as perspectives could be related to identifying amino acids under negative selection within NFC or specifically selected in individual lineages of NFC.

L354-363: *'Besides these, there may still be additional residues involved in the adaptation of NIN for symbiosis, since mutating residues M1 to M7 in Medicago MtNIN still allows the formation of some pink nodules (extended data fig 8). Such additional residues essential for symbiotic NIN functioning might be for example under negative selection, or selected independently in different lineages of the NFC.*

Amino acid changes such as AtNLP2^{M4} improve the symbiotic functionality of AtNLP2 to a level similar to that of PanNIN, enabling rhizobial release and symbiosome formation, but not full bacterial elongation, and leading to premature senescence. The lack of full complementation of the Medicago nin knockout mutant by both modified AtNLP2 and PanNIN reflects a NIN-controlled response evolved later in the legume lineage, after the initial recruitment of NIN in nodulation.'

In all of our complementation experiments, we used the native Medicago NIN promoter, which fully complements the *Mtnin* mutant when driving the Medicago NIN protein. Since this promoter already achieves full complementation, further improvement of the promoter is not needed.

33) Sup Fig 4, the letters indicating the RWP-RK and PB1 domains within the figure are too small, either increase their size, or use a color code (with clear-cut color differences).

> We adapted the figure as suggested

34) The Sup Fig 7 is difficult to use, but I do not really see how to improve it. Please consider again if it is really useful and essential, or not.

> The difficulty to use Sup Fig 7 is inherent to the amount of data it harbours. We have adapted it slightly with additional labelling (including horizontal bars, indicating symbiotic NIN, non-symbiotic NIN, and NIN paralogs) to make it easier to read, also in response to reviewer #1. It is essential to show the quality of the alignment, and it serves as a

references for the readers who want to have a precise view on the data underlying our analysis.

Reviewer #3 (Remarks to the Author):

In this manuscript entitled "Evolution of a NIN-LIKE PROTEIN into a master regulator of Nitrogen-fixing Nodulation" Liu et al. studied the neofunctionalization of the transcription factor NIN for nodulation. First, they tested the ability of NIN orthologs from diverse species to complement nodulation in the the *Medicago* nin mutant. They found that infection thread formation, and in one case organogenesis, can be recovered even with NIN from outside the nitrogen-fixing clade (NFC). The authors used this knowledge to compare sequences and identify that the C-terminal DNA-binding domain is not discriminating between complementing and non-complementing NIN orthologs. By contrast, they found that the increased nuclear localization of NIN was important for its symbiotic function, and linked with putative NLS and several mutations. Studying the evolution of these specific residues, the authors found that their gain predates the evolution of nodulation. Altogether, this suggests that an ancestral NIN, predating nodulation, was already fully functional for its recruitment in a symbiotic context, which is the main conclusion of the manuscript.

The study is based on very solid data: the complementation assays are conducted in the best possible way and scoring all relevant phenotypes (ITs, organogenesis, bacterial release...), the nuclear-localization assays are done in situ, and the EMSA include the appropriate controls. There is a lot of work presented here.

My only major comment is regarding the title. I find it not aligned with the conclusion. "Evolution.. into.." suggests a transition from an ancestral,

non-symbiotic, state to a symbiotic one at the onset of nodulation. As the authors show, NIN was actually already symbiotically-functional before nodulation. The title should be rephrased in a way that reflects the conclusion. Understanding how NIN became integrated into its nodulation-related function will require further work, but this study significantly advance the field in that key direction!

> We thank Reviewer for the positive evaluation of our work and for the thoughtful summary of our findings.

We agree that the original title could be misleading and we have revised the title to better align with our findings:

Ancestral Functionality and Symbiotic Refinement of NIN in Root Nodule Symbiosis.

Response to reviewer comments

Reviewer #1 (Remarks to the Author):

The explanations are acceptable, and the issue has been strengthened in the revised manuscript.

We thank the reviewer for their feedback.

Reviewer #2 (Remarks to the Author):

The revised manuscript now entitled “Ancestral functionality and symbiotic refinement of NIN in root nodule symbiosis” was greatly improved and answered all my concerns. A few minor corrections are still needed before publication.

1) About the title: I found the new title not catchy enough regarding the interest of the study. Maybe something like “Molecular improvement of a non-legume NIN-LIKE PROTEIN into a Symbiotic Regulator of Nitrogen-Fixing Nodulation” could be more clear and attractive?

We thank the reviewer for the suggestion. As previously point out by reviewer #3, we adjusted the title to avoid potentially misleading implications of a state transition. We believe that the current title more accurately reflects the conclusions and the data in the manuscript. Therefore, we decided not to change the title.

2) In the abstract l. 28, “making it functionally comparable to a symbiotic NIN” is too vague and may be misleading for readers, for example by implying that it is as efficient as a complementation of the nin mutant by MtNIN. So please, be more specific. Maybe this could be done just by adding “to the Parasponia PanNIN symbiotic NIN” in the sentence?

We changed the text into:

A single amino acid substitution in the non-symbiotic Arabidopsis AtNLP2 enhances its nuclear localization under low nitrate conditions, making it functionally comparable to the symbiotic Parasponia PanNIN.

3) The Figure 1a seems still to small to me, please increase the size of the letters?

We increased the size of the letters.

4) L. 150, “attached to the GR” sounds strange to me, please consider to modify.

The text now reads “we generated constructs in which the N-terminus of MtNIN was fused to the C-terminus of the tested NIN orthologs and the glucocorticoid receptor (GR)”.

5) In the Figure 2b, it is unfortunate that MeNIN2 does not lead to a significant difference on the NF-YA1 regulation, as this does not fit with the text l. 154. I know that in the Extended Data Figure 4 all other NIN target genes are significantly upregulated by MeNIN2, but this is somewhat confusing, so please consider to show in the main Figure 2b another target than NF-YA1 that would exactly fit the what is said in the sentence l. 154.

To fits better the results in Figure 2 we changed the text, it now reads: “as well as by chimeras containing the non-symbiotic MeNIN1, AtNLP2, and SININ C-termini”.

6) l. 162, replace “constructs” by “chimeric proteins” (this was one of my previous requests already, which may have been not clear enough to you though).

We have changed the text as suggested.

7) In the Figure 2d legend, please indicate that “the order of images from left to right reflects the phylogenic distance to Medicago”. This could (should?) be also mentioned in the other figures where this ordering is used. On a related line, mention in the text of the results l. 170 that “the degree of complementation does not correlate with evolutionary distance”, which to me seems an interesting observation. L. 171, the “,” should be removed.

We have changed the text as suggested.

8) Following my request, you changed the statistical test used in the Extended Data Figure 5, but the legend still reads “Student’s t test”! Please correct, and double check all figure legends in order to correct potential similar mistakes about the statistical tests cited.

We have corrected the legend, it now reads “Student’s t-test on log-transformed data”.

9) Thanks for the extensive modification of the discussion that is now very nice!

We thank the reviewer for their feedback.

Reviewer #3 (Remarks to the Author):

My main concern was regarding the title. The edited version reflects the conclusion and the data presented in the manuscript.

I do not have additional comments.

We thank the reviewer for their feedback.